# Coupling chromosome organization to genome segregation in Archaea

Azhar F. Kabli [1], Irene W. Ng[1], Nicholas Read[1], Parul Pal[2], Julia Reimann[3], Ngat T. Tran[2], Sonja-Verena Albers [3], Tung B. K. Le [2] & Daniela Barillà [1] ✉

Chromosome segregation is a fundamental process in all life forms and requires coordination with genome organization, replication and cell division. The mechanism that mediates chromosome segregation in archaea remains enigmatic. Previously, we identified two proteins, SegA and SegB, which form a minimalist chromosome partition machine in Sulfolobales. Here we uncover patterns and mechanisms that SegAB employ to link chromosome organization to genome segregation. Deletion of the genes causes growth and chromosome partition defects. ChIP-seq investigations reveal that SegB binds to multiple sites scattered across the chromosome, but mainly localised close to the *segAB* locus in most of the examined archaeal genera. The sites are predominantly present in intragenic regions and enriched in one of the two compartments into which the chromosome folds. We show that SegB coalesces into multiple foci through the nucleoid, exhibiting a biased localisation towards the cell periphery, which hints at potential tethers to the cell membrane. Atomic force microscopy experiments disclose short-range DNA compaction and long-range looping of distant sites by SegB, pointing to a significant role for SegB in chromosome condensation that in turn enables genome segregation. Collectively, our data put forward SegAB as important players in bridging chromosome organization to genome segregation in archaea.

Chromosome segregation is a fundamental biological process to ensure the preservation of the genetic material in all life forms. Chromosome replication and segregation occur in separate phases of the cell cycle in eukaryotes and the molecular events underpinning chromosome segregation during mitosis have been investigated extensively in eukaryotes[1].

In bacteria, only a few key players involved in chromosome segregation have been characterised and most commonly include the ParAB system. ParA is a Walker ATPase that binds nonspecific DNA in the presence of ATP and is conserved throughout bacteria and archaea. ParB is a dimeric site-specific DNA-binding protein that recognises the *parS* centromere using a helix-turn-helix (HTH) fold. Several *parS* sites are clustered around the origin of replication[2]. Upon binding to *parS*,

ParB diffuses to adjacent DNA regions that span hundreds of base pairs, forming large nucleoprotein complexes. Another key chromosome segregation protein, a eukaryotic condensin orthologue, known as the Structural Maintenance of Chromosome (SMC) protein, is recruited by ParB to *parS* sites. The bacterial SMC protein forms a rod-shaped homodimer that harbours two helical arms and acts as a molecular 'staple', aligning the two chromosome arms while extruding loops and translocating from origin to terminus region[3–7]. Chromosome replication and segregation are concurrent processes in bacteria.

In stark contrast with eukaryotes and bacteria, knowledge on chromosome segregation in archaea is very rudimentary. The cell cycle of members belonging to different archaeal lineages shows large variability. Genera of the Thermoproteota phylum are characterised by a

[1]Department of Biology, University of York, York, United Kingdom. [2]Department of Molecular Microbiology, John Innes Centre, Norwich, United Kingdom. [3]Molecular Biology of Archaea, Microbiology, Faculty of Biology, University of Freiburg, Freiburg, Germany. ✉e-mail: daniela.barilla@york.ac.uk

short replication stage followed by a prolonged G2 phase, during which the two chromosomes remain aligned and paired while undergoing reorganisation prior to segregation[8,9]. Recent chromosome conformation capture studies have revealed that the chromosome of Thermoproteota members is organised into two spatially segregated compartments, designated as A and B in analogy to chromosome spatial patterns observed in eukaryotes[10]. The A compartment is characterised by high transcriptional activity in contrast to the lower gene expression of the B compartment. ClsN, a protein of the SMC family, associates mostly with regions of the B compartment, mediating its organisation[10]. To date, it remains elusive whether another protein binds and actively establishes the structure of the A compartment.

Previous work by our group identified two proteins, SegA and SegB, that play important roles in chromosome segregation in the thermophilic archaeon *Saccharolobus solfataricus*[11]. SegA is an orthologue of bacterial ParA ATPase proteins, whereas SegB is a site-specific DNA-binding protein that lacks sequence identity to either bacterial or eukaryotic proteins. SegB recognises two imperfect palindromic motifs in the region upstream of the *segAB* cassette, using a ribbon-helix-helix (RHH) DNA-binding domain[11,12]. Whether SegB binds to other sites on the chromosome is not known. SegA binds DNA in a sequence-independent manner[12]. Interestingly, the *segAB* genes are expressed early in M-G1 and S phase and subsequently are downregulated by the cell-cycle regulator aCcr1[13–16]. How the SegAB proteins mediate chromosome segregation is not known.

Here we show that deletion of *segB* and *segAB* causes growth and chromosome segregation defects, resulting in anucleate cells as well as cells that harbour guillotined and multi-lobed fragmented nucleoids. Chromatin immunoprecipitation shows that SegB binds to multiple sites scattered across the chromosome, with a cluster of highly enriched peaks in proximity to the *segAB* locus. Most of the SegB sites are located in the A compartment. Consistent with the ChIP-seq finding, microscopy studies indicate that SegB forms multiple foci by binding to different sites on the chromosome and these foci then coalesce into larger patches, which suggests a role in chromosome organisation. In contrast, SegA is homogeneously distributed throughout the nucleoid. Moreover, atomic force microscopy provides mechanistic insights into how SegB binds DNA, uncovering short- and long-range activities of the protein in DNA condensation. Overall, our findings establish SegAB as key players in chromosome segregation in Thermoproteota and shed light on how the role of these proteins is likely coupled to their intrinsic ability to organise and compact chromosomal DNA.

## Results

### The *segAB* locus and genomic neighbour conservation
The *segAB* genes are found widely on chromosomes of Thermoproteota and Euryarchaeota. The 3' end of *segA* overlaps with the 5' end of *segB* in most genera, suggesting that the genes form a single transcriptional unit. This was verified by PCR on cDNA in *S. solfataricus* (Supplementary Fig. 1). This finding is consistent with previous transcription profiling studies in *Sulfolobus acidocaldarius*[13] and *Saccharolobus islandicus*[14–16] showing that the *segAB* genes are simultaneously expressed in a cell-cycle coordinated fashion.

BLASTP homology searches using *S. solfataricus* SegB as a query identified numerous orthologues that clustered into two well-defined phylogenetic clades (Supplementary Fig. 2). One branch included SegB proteins from Euryarchaeota, whereas the other contains orthologues from Thermoproteota, showing a clear phylogenetic split. Analysis of the *segAB* locus genomic context and gene neighbour conservation, using the FlaGs program[17], showed strict gene synteny of the *segA* and *segB* loci across phyla, with the typical overlapping gene arrangement (Fig. 1). No other gene was associated consistently with *segAB*. Gene neighbour conservation and clustering were quite variable across phyla and genera, albeit with some level of conservation within genera. For example, the genes that flank the *segAB* operon were almost

entirely conserved in *S. solfataricus* and *S. islandicus*. Similarly, members of the *Metallosphaera* genus exhibited highly conserved gene clusters on both sides of the *segAB* locus, including a gene encoding a NAD/FAD-dependent oxidoreductase (cluster 2) that was frequently observed downstream of *segB* also in other genera (*Acidianus* and *Sulfuracidifex*), and genes located upstream of *segA* that encoded Fe-S-binding protein and FAD-linked oxidase proteins (cluster 3 and 4). Interestingly, the genera *Sulfurisphaera*, *Stygiolobus* and *Sulfolobus* shared virtually identical gene clusters upstream (cluster 3, 4 and 11) and downstream (cluster 21, 20, 19 and 25) of *segAB*. A divergently transcribed gene upstream of *segA* (cluster 10) was present in both the *Saccharolobus* and *Metallosphaera* genera. We have recently solved the structure of the encoded protein from *S. solfataricus*, (previously Sso0033, now SegC)[18]. The protein contains a thiamine-diphosphate binding fold and binds phosphate-rich substrates. Notably, SegC interacts with SegA and SegB and the association with SegA results in the formation of polymers in vitro. Thus, the gene synteny underscores a functional relationship that links these three proteins.

### Larger SegB orthologues contain repeats that fold into a sizeable β-helix structure
SegA orthologues are quite uniform in length, ranging from 219 to 261 residues, with the Euryarchaeota proteins being on average ~40 amino acids longer than the Thermoproteota counterparts. In contrast, SegB proteins show a considerable variability in length, fluctuating from 94 to 300 residues (Fig. 1). Inspection of these protein sequences revealed direct tandem repeats (14 amino acids) in the larger SegB of the *Metallosphaera* genus (Supplementary Fig. 3). The sequences, lengths and frequencies of the repeats vary in the different orthologues. Generally, repeats arise from gene duplication events caused by strand mispairing during DNA replication or erroneous repair of double strand DNA breaks[19]. *Metallosphaera hakonensis* harbours the largest SegB (300 residues) that contains 12.5 identical repeats (Supplementary Fig. 3). Whilst the small *S. solfataricus* SegB monomer (109 residues) displays an unstructured N-terminus and a RHH folded C-terminus[12], *Metallosphaera* larger SegB monomers harbour an additional segment inserted between these domains. AlphaFold 2 (AF2)[20] structure prediction indicated with high to very high confidence that the extra domain contains all the repeats and folds into a β-helix in which two extended antiparallel β-sheets are stacked against one another (Supplementary Fig. 4b, Supplementary Data 1 and 2). Each repeat comprises a loop and the majority of a β-strand that straddles two adjacent repeats (Supplementary Fig. 4a). The predicted β-helix comprises six complete turns and displays glutamine ladders at both ends of the β-strands. Glutamines may act as polar zippers in β-helix proteins[21].

RHH proteins function as dimers, in which the two β-strands of the monomers are arranged in an antiparallel fashion to form a β-ribbon that is inserted into the DNA major groove[12]. AF2 multimer[22] was used to predict the structure of *M. hakonensis* SegB dimer (Supplementary Data 3). In the proposed best model, the β-helix is no longer observed. Instead, each monomer presents a flat β-sheet that consists of antiparallel β-strands, although the level of confidence for this domain in the dimer is low. The β-sheet of one monomer is stacked against that of the other monomer (Supplementary Fig. 4c, left). However, one of the five computed models displays the repeat region of each monomer folded into a β-helix as observed in the monomer prediction (Supplementary Data 4). The two β-helices, one from each monomer, sit next to one another in this predicted dimer, merging into a longer solenoid (Supplementary Fig. 4c, middle). Interestingly, the RHH C-terminal domain is correctly folded and predicted with high confidence in both dimer models (Supplementary Fig. 4c). Superimposition of the predicted *M. hakonensis* and the determined *S. solfataricus* RHH domains shows a root mean square deviation (RMSD) of 1 Å, indicating that the structures of these

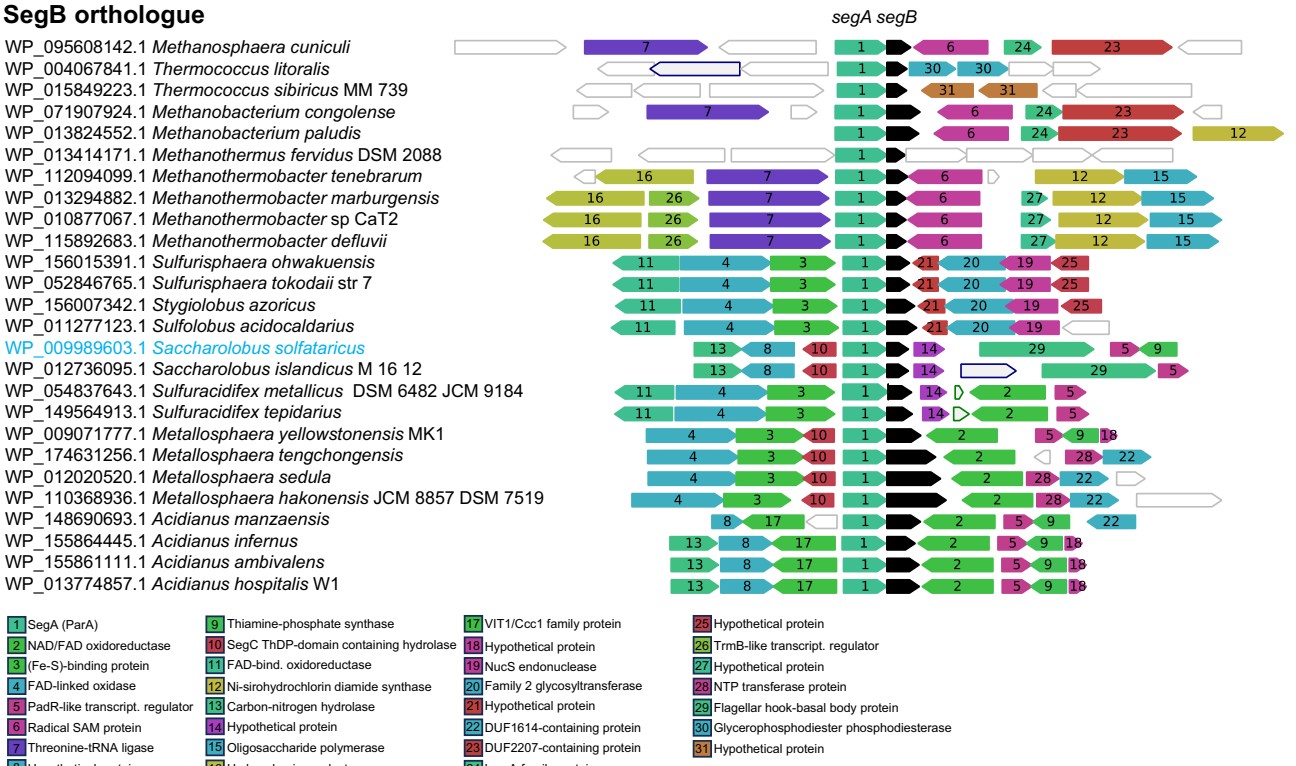

**Fig. 1 | SegB orthologues and genomic neighbourhood conservation of the *segB* locus in archaeal genera encoding the gene.** The webFlaGs programme[17] was deployed to identify SegB orthologues and to investigate the genomic context of the *segB* gene in different archaeal genera. Using *S. solfataricus* SegB sequence as a query, the software performed a BLASTP search against the microbial RefSeq database to identify orthologues using an E value cut-off of 1e⁻3. FlaGs clustered proteins encoded by flanking genes based on homology, using Hidden Markov Model-based method Jackhmmer that is part of the HMMER package[73]. The clustering was performed using an E value cut-off of 1e⁻10, three Jackhmmer iterations and a maximum of four upstream and downstream gene neighbours. Each cluster was assigned a number, starting from 1. The lower the cluster number the more frequently that protein is present among the flanking genes. The black arrow represents the *segB* gene and the other coloured arrows correspond to conserved flanking genes. Non-conserved genes are drawn as white arrows with no number. Arrows with a green outline represent RNA-coding genes and those with a blue outline indicate pseudogenes. The genes and intergenic spaces are drawn to scale. SegB orthologues' accession numbers shown on the left and diagram of the corresponding *segB* gene neighbours for each archaeal species shown on the right. The protein function for each cluster is shown at the bottom using the same colour code deployed for the gene arrows in the neighbourhood diagram. The full output of the FlaGs analysis with all proteins' accession number can be found in the Source Data file.

domains are very similar. AF3[23] also was used to predict the structure of the SegB dimer. The best model shows the N-termini of the monomers folded into adjacent β-helices positioned next to one another, as observed in the AF2-predicted dimer (Supplementary Fig. 4 d and Supplementary Data 5), although with a slightly higher confidence (predicted Template Modelling, pTM = 0.4; interface predicted Template Modelling, ipTM = 0.35). The predicted C-terminus is folded again with high confidence.

Although the function of the β-helix in SegB of *Metallosphaera* is unknown, in Dali[24] and FoldSeek[25] searches using the predicted monomeric SegB structure, the most recurrent hits were the FeS cluster proteins SufB and SufD from bacteria and archaea (Z-score 18.5 for *E. coli* SufB). Interestingly, *E. coli* SufBD form a heterodimer[26] in which the β-helix of each protein sits next to that of the other protein, resulting into an elongated solenoid structure that closely resembles that of the predicted *M. hakonsensis* SegB dimer (Supplementary Fig. 4e, f). Other top hits were a curlin associated repeat-containing protein (Z-score 20.8), S-layer proteins (Z-score 10.8) and various cell surface-associated proteins, in addition to *S. solfataricus* SegB (Z-score 9.5). More distant structural orthologues included bactofilin (Z-score 3.6), a bacterial cytoskeletal protein that harbours a slightly different β-helix fold. Very similar hits were returned, when the structural comparisons were performed using *M. hakonensis* SegB predicted dimeric structure harbouring the β-helices.

## Deletion of *segAB* causes growth and chromosome segregation defects

Gene deletion strains were constructed to establish the role played by the SegAB complex in the cell. Previous overexpression studies suggested a link to chromosome segregation[11]. Deletions were verified by PCR and whole genome sequencing that clearly showed that each deletion strain harbours only the chromosome without the target gene or genes (Supplementary Fig. 5). We were able to construct *S. solfataricus* Δ*segB* and *Sulfolobus acidocaldarius* Δ*segB* and Δ*segAB* deletion mutants. Attempts to generate Δ*segA* in both genera were unsuccessful. Multiple attempts also were made to construct *S. solfataricus* Δ*segAB* without success. The deletion strains exhibited a growth impairment phenotype that was more severe for the Δ*segB* strains than for Δ*segAB* (Fig. 2a). Reintroduction of *segB* on a plasmid in the *S. solfataricus* deletion strain restored the wild type phenotype. Fluorescence microscopy experiments using 4′,6-diamidino-2-phenylindole (DAPI) revealed a high percentage of anucleate cells in all deletion strains, indicating a chromosome segregation defect. *S. solfataricus* Δ*segB* displayed the largest proportion of nucleoid-free cells with 14.93% as compared to 1.0% in the wild type strain (Fig. 2b, c). Flow cytometry experiments also revealed numerous polyploid cells, 45% compared to 26% of the wild type (Supplementary Fig. 6). Interestingly, rescuing Δ*segB* with the expression of *segB* from the plasmid reduced polyploidy to a level even lower than that of the wild type (3%) (Supplementary Fig. 6).

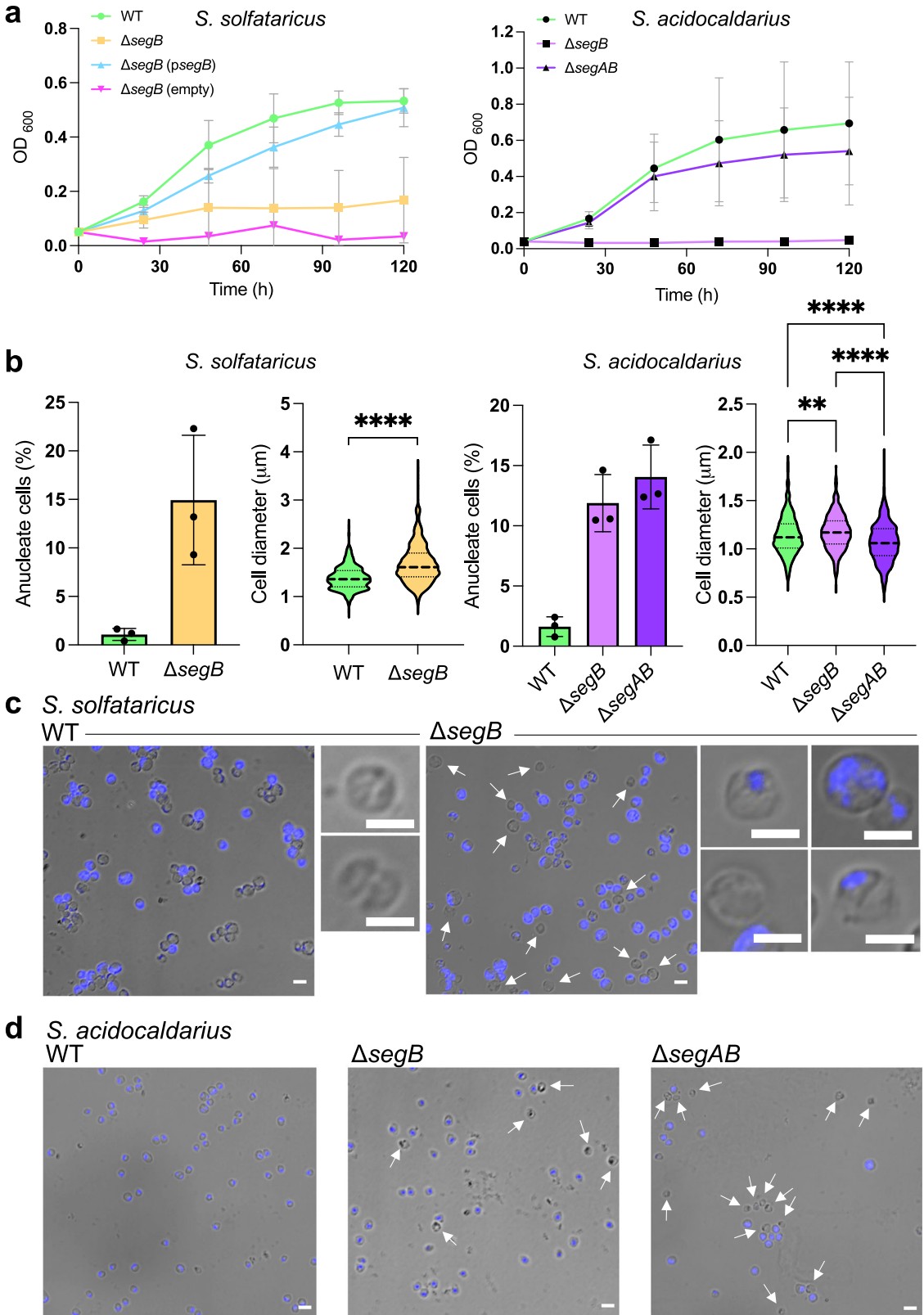

Moreover, aberrant chromosome morphologies and locations were observed in *S. solfataricus* Δ*segB*: some cells had split, fragmented nucleoids, whereas others contained guillotined chromosomes trapped across the division septum, in addition to dividing cells with the nucleoid enclosed in one half of the total area (Fig. 2c). Interestingly, whilst the chromosome assumes a crescent shape compressed beneath the cell membrane in many wild type cells (as

previously reported[8,10]), the chromosomal DNA appears instead more diffuse in the Δ*segB* strain (Fig. 2c).

*S. solfataricus* Δ*segB* cells showed a wide size distribution (0.8 to 3.6 μm diameter) and, overall, were larger (mean diameter 1.68 μm) than wild type cells (mean diameter 1.38 μm) (Fig. 2b). This feature was confirmed by flow cytometry experiments in which *S. solfataricus* Δ*segB* displayed a more heterogeneous population characterised by a

**Fig. 2 | Deletion of *segB* and *segAB* cause growth and chromosome segregation defects. a** Growth curves of wild type, deletion and complemented strains of *S. solfataricus* (left) and *S. acidocaldarius* (right). The plots show the results of three independent biological experiments in which the mean and standard deviation for each time point are indicated (source data are provided as a Source Data file). **b** Plots showing the percentage of anucleate cells and the cell size distribution for wild type and deletion strains of *S. solfataricus* (left) and *S. acidocaldarius* (right) obtained from analysis of microscopy images. Cells were stained with 4′, 6′-dia-midino-2-phenylindole (DAPI) and cell diameters were measured along the longest length. For anucleate cell count, *S. solfataricus* wild type (PBL2025) cells ($n = 798$, mean = 1.080% ± 0.618) and Δ*segB* ($n = 1121$, mean = 14.930% ± 6.671); *S. acidocaldarius* wild type (MW001) ($n = 1062$, mean = 1.623% ± 0.817), Δ*segB* ($n = 1324$, mean = 11.89% ± 2.373) and Δ*segAB* ($n = 565$, mean = 14.06% ± 2.656). The values are mean ± standard deviation and refer to three independent biological repeats. For the cell diameter distribution, *S. solfataricus* wild type (PBL2025) ($n = 432$, mean = 1.387 ± 0.244) and Δ*segB* ($n = 624$, mean = 1.681 ± 0.402) were analysed; *S. acidocaldarius* wild type ($n = 558$, mean = 1.144 ± 0.206), Δ*segB* ($n = 378$, mean = 1.174 ± 0.186) and Δ*segAB* ($n = 528$, mean = 1.069 ± 0.206) were analysed. The data from three biological repeats were combined and plotted in violin graphs in which the bold dashed line indicates the median and the thin dashed lines below and above represent the 1st and 3rd quartile, respectively. Asterisks indicate statistically significant differences: *S. solfataricus* data were subjected to a two-tailed Wilcoxon test (**** $P$ value < 0.0001) and *S. acidocaldarius* data were subjected to the one-tailed Kruskal-Wallis test followed by Dunnett's multiple comparison test (** $p = 0.0074$ for wild type *versus* Δ*segB*; **** $p < 0.0001$ for wild type *versus* Δ*segAB*; **** $p < 0.0001$ for Δ*segB* versus Δ*segAB*). Source data are provided as the Source Data file. **c** Phase contrast and fluorescence microscopy of DAPI-stained *S. solfataricus* wild type and mutant cells, including examples of aberrant chromosome segregation phenotypes, such as guillotined and fragmented nucleoids. Anucleate cells are indicated by white arrows. Scale bar = 2 µm. **d** Phase contrast and fluorescence microscopy of DAPI-stained *S. acidocaldarius* wild type and mutant cells. Scale bar = 2 µm. For both *S. solfataricus* and *S. acidocaldarius* strains experiments were performed at least in triplicates with similar results each time.

broader range of cell sizes and complexities reflective of polyploidy compared to the wild type strain (Supplementary Fig. 7). Syncytia-like assemblies consisting of 3-4 conjoined cells also were noted (Supplementary Fig. 8), suggesting cell separation perturbations. It has been shown in *S. acidocaldarius* that defective chromosome segregation leads to inhibition of cell division[27].

*S. acidocaldarius* Δ*segB* and Δ*segAB* populations contained 11.89% and 14.06% of anucleate cells, respectively, in contrast to 1.62% for the isogenic wild-type strain (Fig. 2b). Size distributions exhibited some significant differences between mutant and the wild type strains, although cell diameters were not very different (Fig. 2b). Nucleoids appeared well defined in Δ*segB* cells, mostly at midcell, whereas nucleoids were more diffuse and broader in Δ*segAB* cells (Fig. 2d). Unseparated cell clusters and aberrantly shaped cells were observed for the Δ*segAB* strain, some displaying bleb-like protrusions (Supplementary Fig. 8). Flow cytometry experiments also disclosed polyploid cells in both deletion mutants (Supplementary Fig. 6). Overall, the results indicate that the SegAB proteins are involved in chromosome segregation and that the disruption of this process may block downstream cell cycle events, including cell division.

## SegB binds to multiple sites scattered across the *S. solfataricus* chromosome and recognises a specific motif in vivo

Previous investigation established that *S. solfataricus* SegB binds to an imperfect palindromic motif that is located just upstream of the *segA* start codon (site 1) and further upstream, centred at position -59 from the same codon (site 2)[11]. A genome-wide map of the sites recognised by *S. solfataricus* SegB in vivo was compiled by chromatin immunoprecipitation (ChIP) with anti-SegB antibodies followed by sequencing, using cells from wild-type *S. solfataricus* P2, PBL2025 strains and Δ*segB* deletion strain. Sequences of the immunoprecipitated DNA were mapped on to the *S. solfataricus* P2 genome (NC_002754), leading to the identification of pulled down sites. Thirty-nine enrichment peaks were identified across the chromosome (Fig. 3, Supplementary Table 1). Among these, 15 were high- and 24 low-enrichment peaks, defined as peaks with a RPBPM value inferior to 5. This result suggests the presence of SegB sites with different binding affinity. The peaks are narrow, indicating that SegB binds to discrete cognate sequences with no long-range spreading.

A cluster of highly enriched peaks is located in the region that encompasses the *segAB* locus and *oriC1* region. Closer inspection of the extended *segAB* area (coordinates 0 to 50 kb) revealed 16 peaks, most of which are high-enrichment and likely to be strong affinity binding sites for SegB. Peaks B1 to B7 are located upstream of the *segAB* locus and another seven clusters in the region that spans *segC* and *segAB* (Fig. 3c). The B10 peak contains the experimentally characterised sites 1 and 2[11]. A major peak (C) and three other smaller peaks

are located further downstream in the extended *oriC1* region (Fig. 3b, d and Supplementary Table 1). A few low-enrichment peaks are observed close to *oriC3* and another cluster that contains multiple peaks, including a high-enrichment peak D, is localised in the intergenic region between SSO_RS05720 and SSO_RS05725 (Fig. 3b, e and Supplementary Table 1). Then, a large chromosomal zone devoid of significant peaks extends up to *oriC2*. A few notable peaks are observed in proximity to this replication origin, including a high-enrichment site (peak E) that is located in the SSO_RS10935 locus (Fig. 3b, f and Supplementary Table 1). This gene encodes the ClsN protein, an atypical SMC-like factor that replaces condensin in Sulfolobales and is enriched in the B compartment of the chromosome[10]. A number of other low-enrichment peaks are present in the remaining arm that stretches until the beginning of the high-enrichment region surrounding the *segAB* locus. The sector that spans between ~1.003 and 1.9 Mb (approximately between peaks D and E) is the most impoverished in SegB binding sites (Fig. 3a). This region falls within the B compartment[28].

A search for de novo SegB DNA-binding motifs within the sequences corresponding to the peaks identified a 13 bp palindromic motif with the consensus sequence 5′-TCTAGA S TCTAGA-3′ (where S is either G or C) (Supplementary Fig. 9). The motif was found in all the sequences corresponding to the 39 enrichment peaks, with some peaks harbouring more than one site. In total, 46 sites containing the motif were identified bioinformatically. The consensus motif aligns extremely well with that previously reported[11]. Most of the enriched sites were found in intragenic regions (83.7%), with only 5 sites in intergenic sequences (Supplementary Fig. 9b).

Although the genes containing SegB binding sites encode proteins with diverse functions (Supplementary Fig. 9c and Supplementary Table 1), the majority encodes for hypothetical proteins. In addition to the previously mentioned peak E that is located within the gene encoding coalescin ClsN, low-enrichment peaks of significant interest are found within the gene SSO_RS01130 (*rpoB2*) and SSO_RS04795 (*topR-2*) that encode the B″ subunit of RNA polymerase and the reverse gyrase, respectively (Supplementary Table 1). Altogether, the identification of multiple SegB binding sites distributed across most of the chromosome and the potential functional linkage to factors and processes involved in chromosome topology and compartmentalisation suggest that SegB may exert a role in chromosome organisation that in turn enables accurate segregation.

## Conserved patterns in genome-wide distribution of SegB binding motif in diverse archaeal genera

The palindromic consensus motif identified for *S. solfataricus* SegB was used to search for a similar motif in the genomes of other archaeal genera that encode SegB orthologues. These searches identified a 13 bp identical motif and variants on the chromosomes of

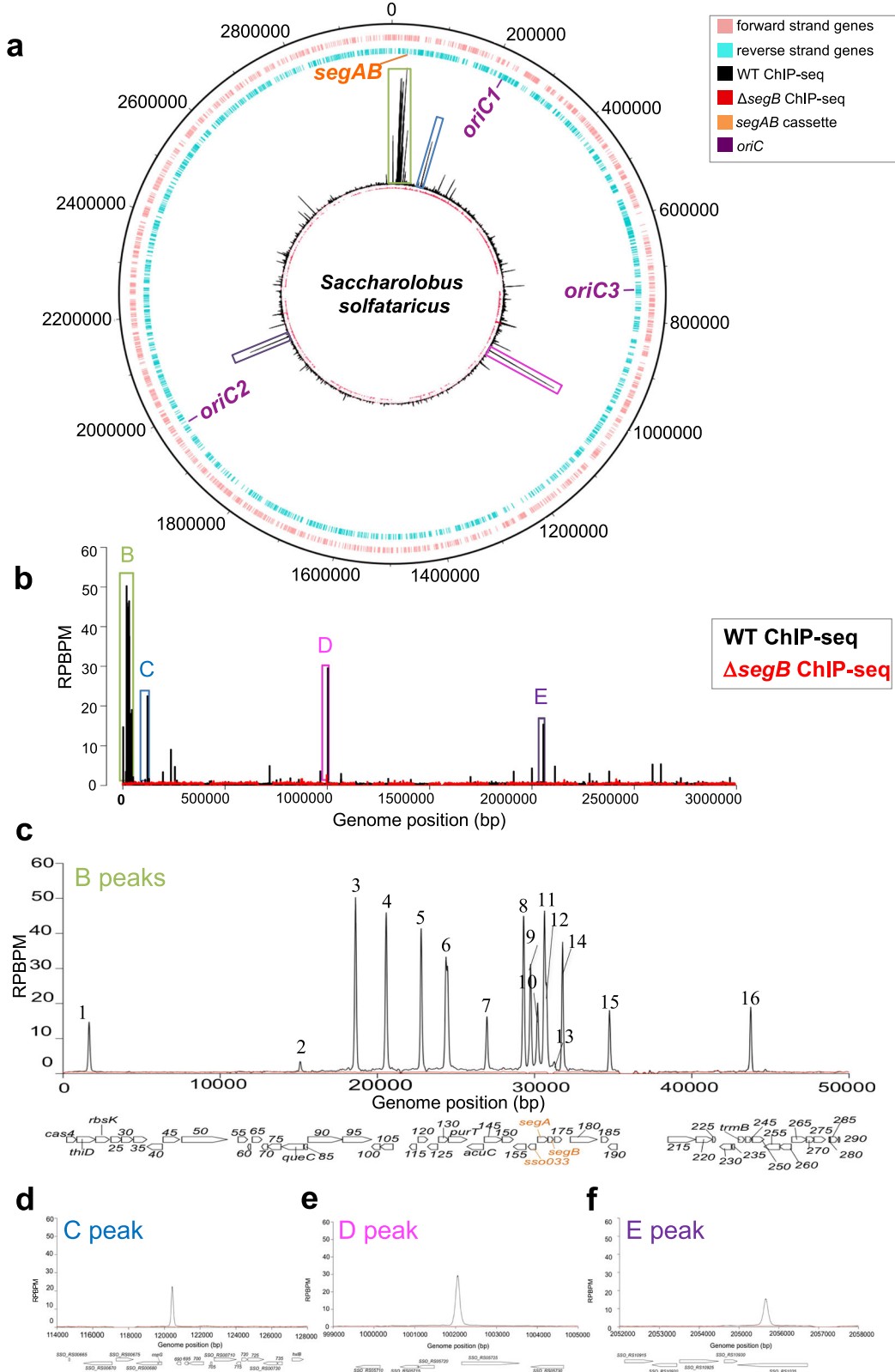

**Fig. 3 | SegB from *S. solfataricus* binds to multiple sites scattered across the chromosome. a** Circular plot of the *S. solfataricus* chromosome showing forward strand genes in pink and reverse strand genes in blue. The position of the *segAB* locus and the three *oriC* is indicated. The internal circles show the ChIP-seq profile of SegB enrichment on *S. solfataricus* (wild type) (black) and Δ*segB* (red) chromosome. High-enrichment peaks are enclosed in coloured boxes adopting the same colour code used in panel (b). **b** Zoomed-in ChIP-seq profile of SegB enrichment on wild type *S. solfataricus* P2 (black) and Δ*segB* (red) chromosome. The Y axis represents the number of reads per base pair per million of mapped reads (RPBPM) at each genomic position (X axis). Major enrichment peaks are enclosed in coloured boxes. Two biological replicates were performed. **c** Zoomed-in image of B peaks clustered around the *segAB* locus in the region spanning from 0 to 50000 bp. The genes encoded in this region are shown below. **d** Zoomed-in image of C peak with genome coordinates. **e** Zoomed-in image of D peak with genome coordinates. **f** Zoomed-in image of E peak with genome coordinates.

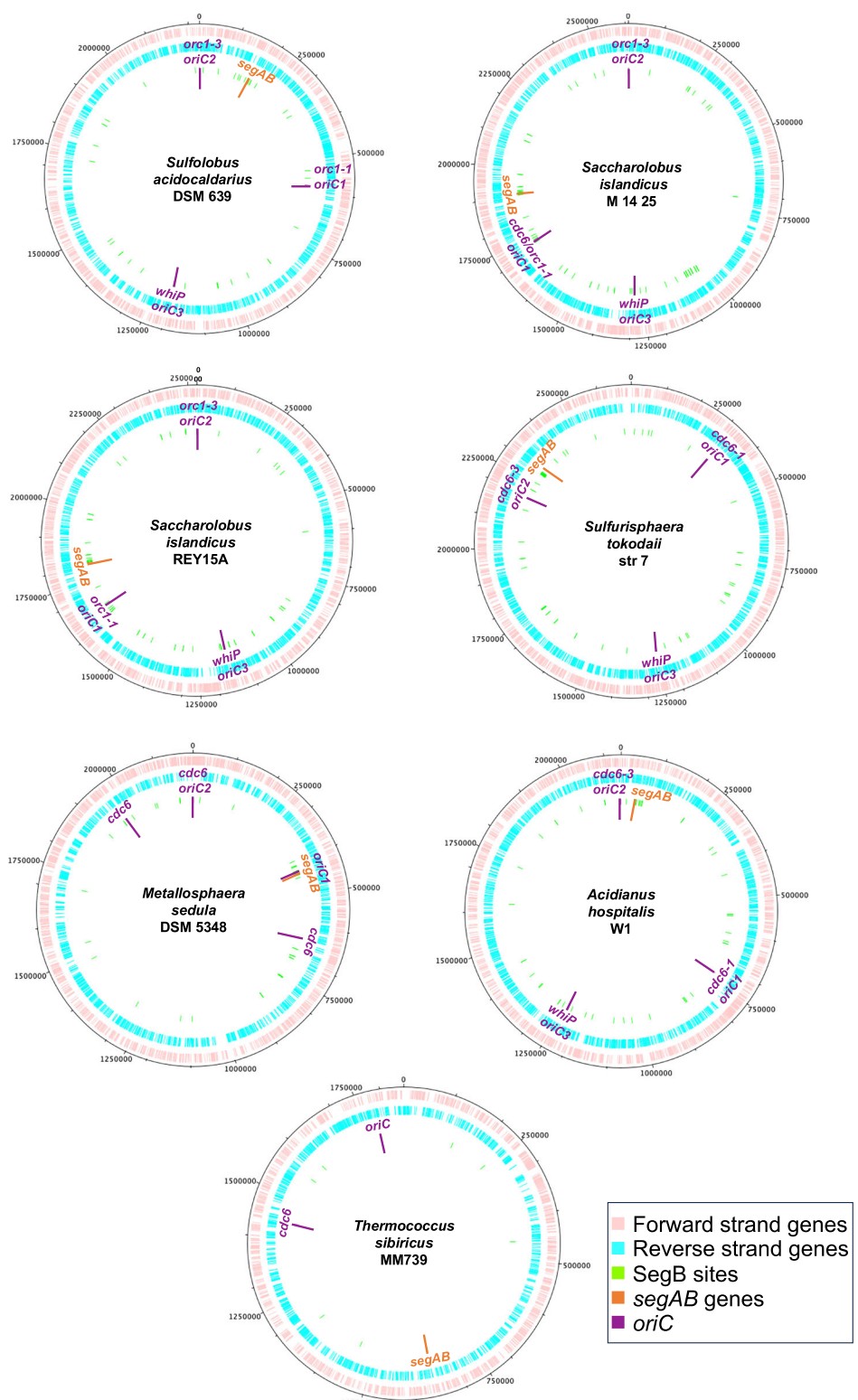

**Fig. 4 | Bioinformatic identification of SegB sites on the chromosome of different archaeal genera.** Circular plots of the chromosomes of different archaeal species showing forward-strand genes in pink and reverse-strand genes in blue. The position of the *segAB* locus and the three *oriC* is indicated. FIMO searches[69] were performed, deploying the *S. solfataricus* SegB motif as a query. The position of the identified binding site is indicated by green lines.

Thermoproteota and Euryarchaeota (Fig. 4). The motif appears to be well conserved across different genera, with an almost invariant central core 5′-AGA g/c TCT-3′.

As observed in *S. solfataricus*, several SegB binding sites are closely clustered in proximity to the *segAB* locus, whereas other sites are spread over different chromosomal locations. The sites containing the most conserved consensus sequence are located close to the *segAB* genes, suggesting that these sequences may be key high-affinity sites. The chromosomal region between *oriC2* and *oriC3* in Sulfolobales appears on average, impoverished in SegB sites, as already experimentally observed in *S. solfataricus*. This region corresponds to a segment of the B compartment on the chromosome of *S. islandicus*

REY15A[10] and of *S. solfataricus* P2[28], which suggests that SegB sites are enriched in the A compartment. Interestingly, *oriC1* is located next to the *segAB* locus on the *Metallosphaera sedula* chromosome (Fig. 4).

Bioinformatic analysis uncovered only seven SegB binding sites with high *P* value (between 6.6 and 8.50 1E-06) and none near to *segAB* in the chromosome of *Thermococcus sibiricus*, which suggests that the motif might be more degenerate in Euryarchaeota. The consensus sequence of the identified sites is very similar to that of *S. solfataricus* (TCT AGA a/t TCT ACA). It is likely that other sites that harbour a sequence quite different from the motif observed in Sulfolobaceae are present in proximity to the *segAB* genes and elsewhere on the chromosome. To test this hypothesis, the *P* value threshold for the search was lowered (from 1.0E-5 to 1.0E-4) and 193 sequences were returned, indicating that SegB sites are more degenerate in *Thermococcus sibiricus*. The genomic context of *segAB* is also quite different compared to that of the Thermoproteota genera (Fig. 1). Nevertheless, the conservation of the motif across diverse genera of Thermoproteota points to potential similarities in the binding modes of different SegB orthologues.

## SegB sites are enriched in the A compartment

The bioinformatic searches suggested that SegB binding sites are rare in the B compartment, but enriched in the A compartment. ChIP-seq experiments were performed on *S. acidocaldarius* cells to test this tantalizing hypothesis. Sequences of immunoprecipitated DNA fragments were mapped on to the genome of *S. acidocaldarius* (NC_007181.1), which revealed 32 enrichment peaks across the chromosome (Fig. 5 and Supplementary Table 2). A few of these peaks are high-enrichment and the rest are medium to low-enrichments peaks (defined as peaks with a RPBPM value inferior to 5), which again suggests the likely presence of high-affinity and lower-affinity SegB binding sites. The peaks are sharp and narrow, suggesting that SegB binds to discrete cognate sequences.

A large group of 15 neighbouring peaks clusters in proximity to the *segAB* locus, whereas the others are scattered downstream in the region of *oriC1* and *oriC3*. A few peaks are present beyond *oriC3*, but then a region devoid of SegB sites is apparent, which is consistent with the bioinformatic predictions. This region sits within the chromosomal B compartment[10]. Notably, 81.25% of the enrichment peaks are positioned in the A compartment and only 18.75% is in the B domain (Fig. 5a, d), which suggests that SegB may undertake an important role in organising the A compartment.

De novo searches for SegB binding sites within enriched peaks in *S. acidocaldarius* identified a 13 bp motif with the consensus 5′-STAGASTCTAKAG-3′ (where S is either G or C and K is either T or G) (Supplementary Fig. 9d). This consensus is very similar to that identified above for *S. solfataricus* SegB. The motif was found in all the sequences corresponding to the 32 enrichment peaks for a total of 32 sites. Most of the sites were found in intragenic regions (82.85%) (Supplementary Fig. 9e), as already observed in *S. solfataricus*. Only 6 sites are intergenic and 4 of these reside in the sequence upstream of *segAB*. SegB sites are located within genes that encodes proteins performing diverse functions (Supplementary Fig. 9f and Supplementary Table 2), including within *segAB*. In contrast to *S. solfataricus*, no SegB site is present in the coalescin gene of *S. acidocaldarius*. However, a site in the gene that encodes the nucleoid-associated protein Alba (SACI_RS06315) hints again at a potential functional interplay between SegB and factors involved in chromatin architecture and genome packaging.

## SegB binds in vitro to the sites identified through ChIP-seq

Electrophoretic mobility shift assays (EMSA) were performed with *S. solfataricus* SegB protein to establish whether the sites identified in the ChIP-seq enrichment peaks were indeed bound by the protein. Fragments containing the sequences identified for the high-enrichment peaks B3 to B6 (located upstream of *segAB*) (Fig. 3c) and peak D (located between SSO_RS05720 and SSO_RS05725) (Fig. 3e) were incubated with SegB and the complexes were resolved on gels. Each of the fragments was bound by SegB in the presence of competitor DNA, indicating that the fragments contain one or more specific binding sites for the protein, thereby validating the ChIP-seq results (Supplementary Fig. 10).

The DNA fragments corresponding to different peaks were subjected to DNase I footprinting, which revealed discrete windows of protection. Within them, one or more adjacent SegB binding motifs were identified (Fig. 6). Interestingly, the region that corresponds to peak B-6 showed three distinct windows of protection, within which three SegB sites were recognised. These results reveal that the overall number of SegB binding sites may be higher than the number of peaks apparent in ChIP-seq, as some peaks may harbour more than one site. Forty-nine total sites were identified on the *S. solfataricus* chromosome: the bioinformatic analysis identified forty-five sites. Moreover, DNase I footprints detected four additional sites, bringing the total number of identified sites to forty-nine (Supplementary Table 1).

A 50 bp oligonucleotide that contains the 5′-end of *segA* and upstream region was incubated with purified SegB and subjected to EMSA to corroborate the binding of *S. acidocaldarius* SegB to sequences uncovered by ChIP-seq. SegB bound to this fragment (Supplementary Fig. 11c). In addition, a larger region corresponding to the sequence under peak 15 (Supplementary Fig. 11b) was investigated by DNase I footprint. The bioinformatic search identified a single SegB site within this sequence (5′-CCGTAGTCTGGAG-3′). Interestingly, the footprint showed a large window of protection in which three additional SegB binding sites were uncovered (Supplementary Fig. 11a). Thus, at least 35 SegB sites are present on the *S. acidocaldarius* chromosome based on bioinformatics and in vitro analyses. Overall, the biochemical SegB-DNA investigations confirmed the ChIP-seq results, but also revealed discrete adjacent binding sites for SegB within the sequences of multiple peaks.

## SegA and SegB form diverse structures over the nucleoid that correlate with their distinct DNA-binding features

The subcellular localisation of SegA and SegB was investigated in *S. solfataricus* cells to obtain further insights into the mode of action of the proteins. Immunofluorescence microscopy revealed that the SegA signal is distributed rather uniformly throughout the nucleoid, forming extensive patches, which is consistent with non-sequence-specific DNA-binding by the protein (Fig. 7a). Localisation of the DAPI and SegA signals was well-correlated (Fig. 7b). Similar localisation patterns were established for bacterial ParA orthologues, although these proteins are known to dynamically relocate across the chromosome[2,29,30]. As thermostable fluorescent proteins were not available, it was not possible to assess further the dynamics of SegA. However, Z-stack sections of the cells showed that the SegA signal does not simply coat the chromosome surface, but that the signal diffuses throughout the whole volume of the nucleoid, permeating the chromosome in three dimensions (Fig. 7c).

SegB formed discrete foci of different sizes within the nucleoid (Fig. 7d), which is consistent with ChIP-seq results indicating that the protein binds to multiple chromosomal sites. This interpretation is also supported by the site-specific nature of SegB binding to DNA. Most cells showed multiple foci (Fig. 7di, ii). Some cells contained large SegB patches that likely originated from the coalescence of smaller foci (Fig. 7dii, iii), potentially suggesting that SegB may bridge physically separate, non-adjacent sites. A minority of cells exhibited a single focus/patch (2%), whereas most of the cells harboured five or more foci (54%) (Fig. 7e). Foci were present throughout the nucleoid volume as shown by Z-stack sections (Fig. 7g).

It can be challenging to establish spatial reference points for protein localisation in *S. solfataricus* which is an irregular coccus.

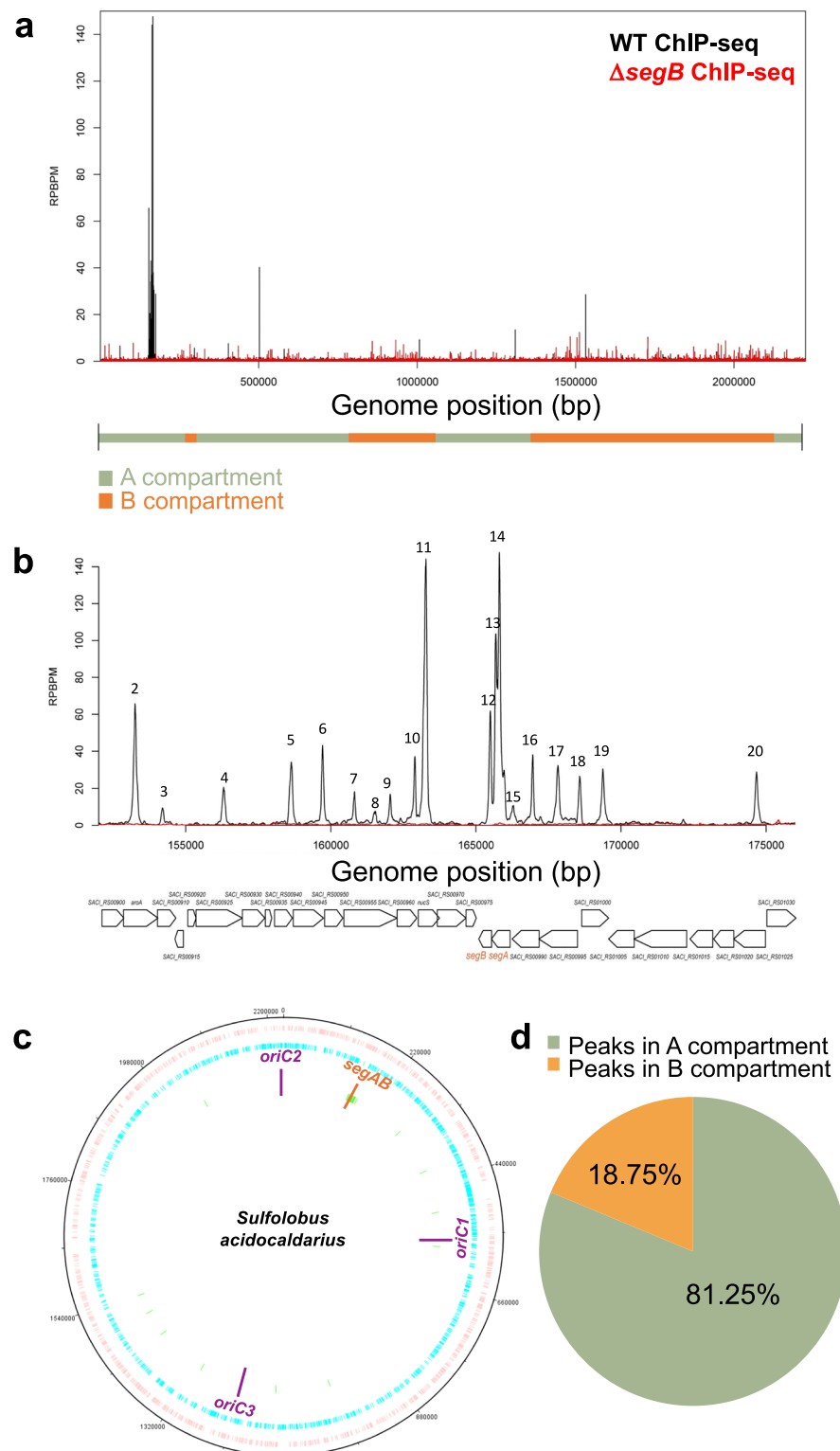

**Fig. 5 | SegB sites on the chromosome of *S. acidocaldarius* are predominantly located within the A compartment. a** ChIP-seq profile of SegB enrichment peaks on the chromosome of wild type *S. acidocaldarius* (black) and ∆*segB* (red). The Y axis shows the number of reads per base pair per million of mapped reads (RPBPM) at each genomic position (X axis). The position of A and B compartments[10] is shown below the X axis. Two biological replicates were performed. **b** Zoomed-in plot of the SegB high-enrichment peaks in the region containing the *segAB* locus, wild type (black) and ∆*segB* (red). The genes encoded in this region are shown below. **c** Circular plot of *S. acidocaldarius* chromosome showing forward strand genes in pink and reverse strand genes in blue. The position of the *segAB* locus and the three *oriC* is indicated. Green lines denote the position of the SegB peaks identified by ChIP-seq. **d** Pie chart showing the percentage of SegB peaks in the A (green) and B (orange) compartment of *S. acidocaldarius* chromosome.

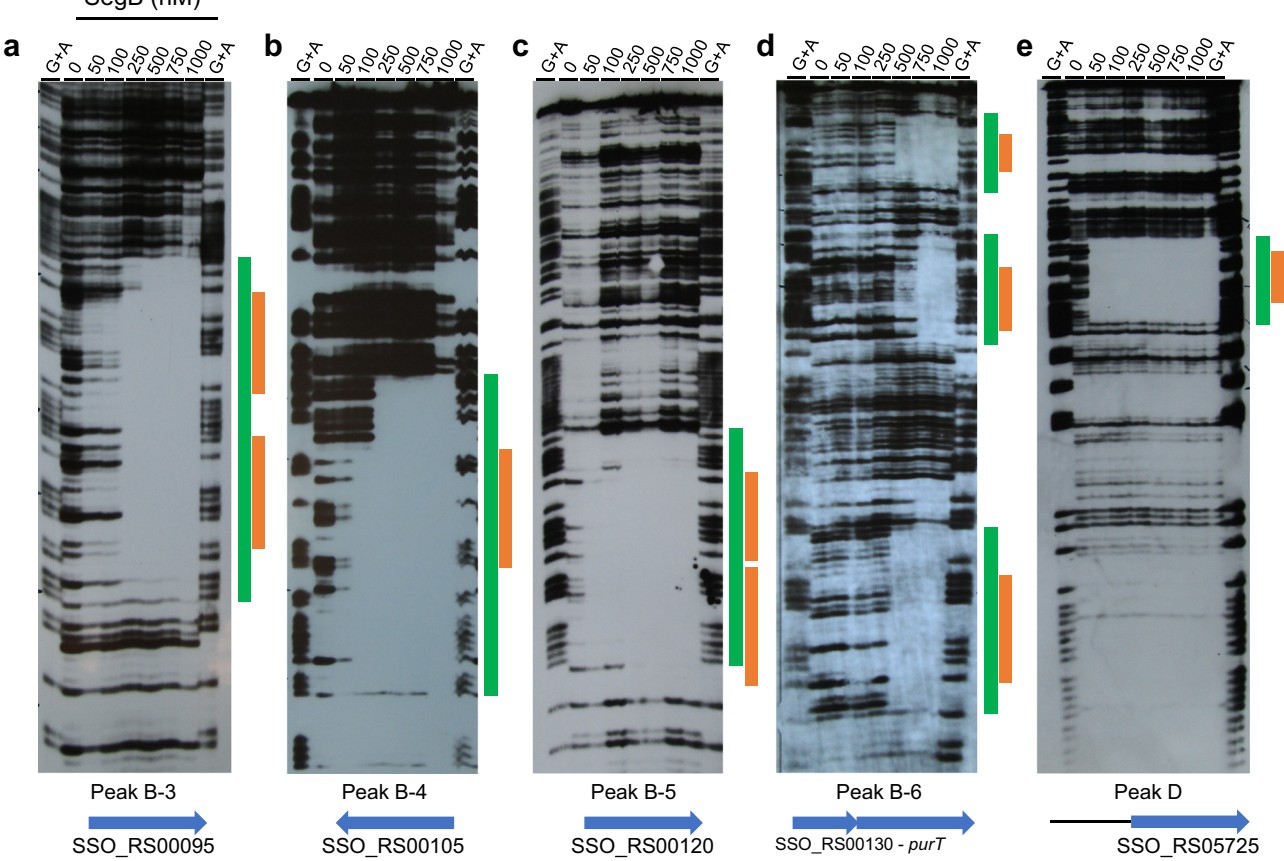

**Fig. 6 | DNase I footprints identify the SegB$_{Ss}$ sites contained within DNA regions corresponding to different peaks.** DNase I footprints showing the window of protection caused by SegB binding to the DNA sequence that encompasses peak B-3 (**a**), peak B-4 (**b**), peak B-5 (**c**), peak B-6 (**d**) and peak D (**e**). The green bar indicates the window of protection and the orange bar denotes the bound site/s. The gene/s present within the peak's sequence are shown at the bottom. Three or two independent experiments were performed for each DNA sequence with similar results.

To examine SegB position, the midcell, cell periphery (cell membrane), and the normalised radius were used as spatial landmarks. Quantitative analysis ($n = 1026$) showed that SegB foci are distributed across the full cell radius in asynchronous cell populations (Fig. 7f). However, the most frequent location on average is between mid-radius (0.5) and slightly past the three-quarter radius position (0.85) in proximity to the cell periphery. The midcell point and the extreme cell periphery are areas of low SegB occupancy. Interestingly, when cells display a single SegB focus/patch, this is most frequently positioned almost at mid-radius (0.4), whereas in cells with four foci the preferred position shifts to 0.6 and the location moves even further towards the cell membrane (0.7-0.8) in cells with more than five foci (Supplementary Fig. 12). Although the significance and mechanism underlying this trend are currently elusive, the distribution pattern may be indicative of potential connections with unknown interacting factors.

**SegA binds SegB with high affinity**

Previous sedimentation and dynamic light scattering studies showed that *S. solfataricus* SegA and SegB form a complex[11]. Here, this interaction was investigated quantitatively by deploying microscale thermophoresis (MST). SegB was fluorescently labelled and mixed with increasing concentrations of SegA. Alterations in the motion or thermophoresis of SegB molecules in a laser-generated temperature gradient were monitored using changes in fluorescence. SegA interacted with SegB with a derived $K_D = 46 \pm 10$ nM, which indicates a strong association between the two proteins (Supplementary Fig. 13).

**SegB binds to higher- and lower-affinity sites leading to DNA compaction in vitro**

The ChIP-seq and DNase I footprint investigations highlighted the presence of higher and lower affinity sites for SegB on the chromosomes of both *S. solfataricus* and *S. acidocaldarius*. Atomic force microscopy (AFM) experiments were performed to obtain mechanistic insights into DNA binding by *S. solfataricus* SegB. The region upstream of the *segAB* locus was selected as it contains three separate SegB sites[11,12] (Fig. 8a). Images of free DNA fragments (289 bp) and DNA-SegB complexes were analysed. Free DNA fragments exhibited uniform height (~0.4 nm) along their length and no localised foci (Fig. 8bi and ii, top left panel, and Fig. 8c, d). When the DNA was incubated with SegB (100 nM), most of the DNA molecules displayed a single bright focus that corresponds to site 1 bound by SegB (Fig. 8bi and ii, top right panel, and Fig. 8c), based on location at the end of the fragment and previous data establishing that this site is the first to be occupied[11]. The focus region had an average height of ~1.6 nm (Fig. 8c, d) compared to ~0.4 nm for the rest of the DNA length. Site 3 and then site 2 were bound by SegB at increased protein concentration (300 and 700 nM) (Fig. 8b bottom left and bottom right panel). This pattern mirrors precisely the progressive occupation of SegB sites observed in previous DNase I assays[11], further supporting the existence of higher and lower affinity sites. Statistical analysis of multiple DNA molecules showed that site 1 remained the most prominent site at high SegB concentration, as evidenced by an average focus height of 1.4 nm compared to ~0.5 - 0.6 nm for the other two foci (Fig. 8d). Interestingly, the focus on site 1 became more spread out at higher SegB concentrations, covering a larger DNA area, suggesting that SegB spreads

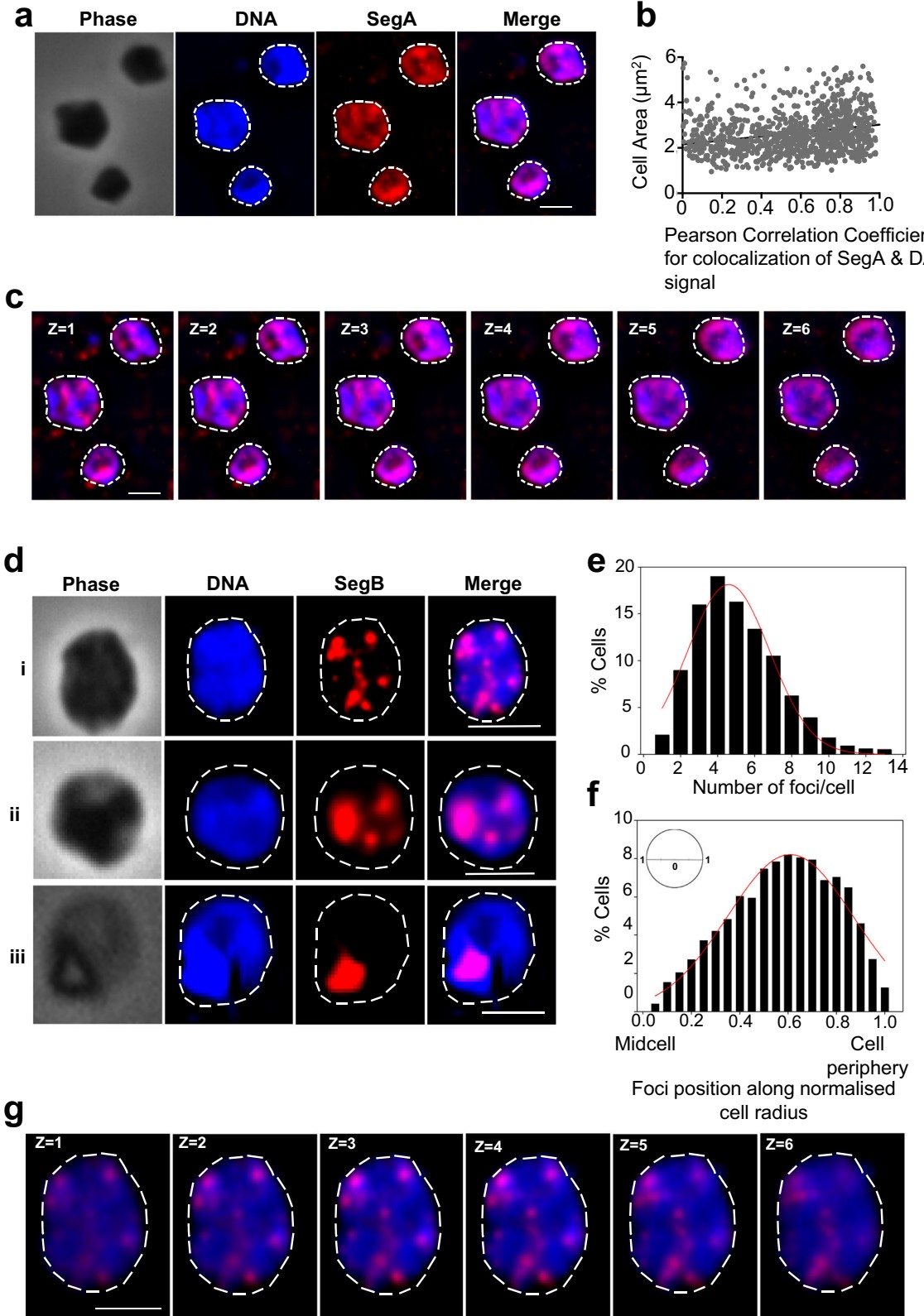

laterally towards the end of the DNA fragment (Fig. 8d). This feature may hold true for site 2 and 3 as well, although it is less obvious due to the substantially lower average heights of the respective foci.

The contour lengths of bare DNA and SegB-bound DNA fragments were measured (Fig. 8e). The theoretical length of the 289 bp DNA fragment is 98.26 nm, which is consistent with the measured average contour length of 94.31 ± 3.56 nm. The average contour length of DNA

molecules presenting a single focus was 6.6% smaller (~6 nm shorter), when the DNA was incubated with SegB (100 nM) (Fig. 8e, top right panel). A more considerable decrease in DNA length was observed at higher SegB concentrations, when the three sites were occupied. The average contour length was 15% (~14 nm shorter) and 18.4% (~17 nm shorter) more compact for fragments incubated with 300 and 700 nm SegB, respectively (Fig. 8e, bottom panels). These data indicate that

**Fig. 7 | Localisation of SegA and SegB in *S. solfataricus* cells. a** Representative immunofluorescence microscopy images showing SegA localisation. PBL2025 cells were immunolabelled using anti-SegA antibody and DNA was stained with DAPI. The white dashed line indicates the cell perimeter. Scale bar = 2 µm. Three independent experiments were performed with similar results. **b** Distribution of SegA and DAPI signal across the cell area (*n* = 1012). Cell area indicated on the Y axis and Pearson Correlation Coefficient on the X axis. **c** Z-stacks of six z-planes taken at 0.2 µm intervals. The white dashed line indicates the cell perimeter. Scale bar = 2 µm. **d** Representative immunofluorescence microscopy images showing SegB

localisation over the nucleoid in cells containing five or more foci (i), four foci and patches (ii) and one single patch (iii). The white dashed line indicates the cell perimeter. Scale bar = 2 µm. Three independent experiments were performed with similar results. **e** Plot showing the number of SegB foci/cell (*n* = 1026). **f** Position of SegB foci along the normalised cell radius (0 corresponds to midcell and 1 to the cell periphery) (*n* = 1026). **g** Z-stacks of six z-planes of the cell in panel D i taken at 0.2 µm intervals. The white dashed line indicates the cell perimeter. Scale bar = 2 µm. For the graphs source data are provided as Source Data file.

SegB binding results in DNA compaction and shortening, revealing a short-range action of this protein in chromosomal DNA organisation. In view of the number of SegB sites scattered across the chromosome, the cumulative effect of this short-range activity may play an important role in chromosome condensation in all the archaeal genera that deploy SegB orthologues.

### SegB mediates DNA loop formation by bridging long-distance sites

As SegB sites are present in different regions of the chromosome, AFM experiments were designed to study the potential bridging of distantly located binding sites. DNA fragments that contain two or three high-affinity sites positioned ~1000 bp apart (~350 nm) were constructed (Fig. 9a). DNA fragments displayed rather linear and open shapes in the absence of SegB (Fig. 9ai, ii, bi, ii). However, DNA loops were observed clearly when SegB was incubated with the DNA fragment that harboured two sites (Fig. 9ai, ii, middle and bottom panels). Moreover, bow-shaped structures exhibiting two conjoined loops were visible when the protein was mixed with a fragment that contained three binding sites spaced by 1000 bp (Fig. 9bi, ii, middle and bottom panels). These results indicate that SegB binds to the cognate sites and then bridges these sites, thereby mediating the formation of loops. This long-range action is underpinned by the ability of SegB to form dimers and dimers-of-dimers, as shown by previous tertiary structure studies[12]. The bridging of non-adjacent regions may promote the coalescence of SegB foci observed in microscopy images. The folding of chromosomal DNA into multiple loops may mediate chromosome compaction and contribute to compartment organisation in the cell. Taken together, the short-range and long-range actions revealed by AFM point to an important role for SegB in chromosome organisation and folding in archaea.

## Discussion

The fundamental process of chromosome segregation remains highly enigmatic in the archaea domain. We have previously identified the SegA and SegB proteins that showed involvement in chromosome partition[11]. Here, we have uncovered mechanisms adopted by these proteins that link chromosome organisation with genome segregation.

The *segAB* genes form an operon (Supplementary Fig. 1). The expression of *segAB* is cell-cycle regulated and begins at the M-G1 transition, reaching a peak in S phase and then declining in G2, due to the repression activity of the aCcr1 cell-cycle transcription factor[14–16]. The aCcr1 repressor has a predicted RHH structural fold[15] that is also present in SegB[12]. Although most transcription factors in archaea contain a helix-turn-helix DNA binding motif[31], the RHH fold is emerging as a recurrent theme in archaeal cell cycle regulator proteins. For example, *Haloferax volcanii* and *Halobacterium salinarum* CdrS harbour a predicted RHH domain and regulate cell division and cell shape[32,33]. The expression of *segAB* also may be subjected to a regulatory feedback loop, as SegB sites are present in the promoter region in *S. solfataricus* and *S. acidocaldarius*, although it remains to be determined whether SegB has a role in transcriptional repression. The presence of SegAB during the M/D/G1 and early S phases suggests that the action of these proteins starts early in the cell cycle prior to and during chromosome replication.

The *segAB* gene deletions cause chromosome segregation defects leading to anucleate and polyploid cells as well as aberrant nucleoid morphology (Fig. 2; Supplementary Fig. 6, 7). Despite the deletion, the mutant strains are viable. It is likely that some level of redundancy exists with respect to the systems and mechanisms responsible for chromosome segregation. In bacteria, the ParAB system is essential for viability only in some species, such as *Caulobacter crescentus*[34]. However, even when the system is not strictly required for viability, deletion of the genes results in significant defects and an increase in anucleate cells, substantiating the pivotal role played in chromosome segregation[35]. Interestingly, in *S. acidocaldarius* the deletion of both genes is less deleterious and better tolerated than the deletion of *segB* alone, despite the marginally higher percentage of anucleate cells. The co-expression of the two genes from the same promoter signals that stoichiometric quantities of the proteins are important for their function in the cell. When an imbalance of the two proteins occurs, then defects arise. For bacterial ParAB systems, equal concentration of the two proteins is key for the DNA segregation function[34,36].

The average cell size of the deletion strains is larger (Fig. 2 and Supplementary Fig. 7) and conjoined cell clusters are also observed (Supplementary Fig. 8), suggesting that perturbation of chromosome partition affects subsequent steps of the cell cycle, including cell division. Defective chromosome segregation results in inhibition of cell division in *S. acidocaldarius*[27]. Similarly, bacterial mutants impaired in chromosome segregation exhibit cell division site selection anomalies, resulting in mis-localisation of cell division proteins, mis-positioning of division septa over the nucleoid and guillotined chromosomes[37–43]. Thermoproteota divide by using an ESCRT (Endosomal Sorting Complex Required for Transport) system[44,45]. Whether the absence of SegA and SegB modulates the localisation of cell division proteins in *S. solfataricus*, *S. acidocaldarius* and other archaea remains to be investigated.

The ChIP-seq analyses revealed multiple SegB binding sites scattered across the chromosome in both *S. solfataricus* and *S. acidocaldarius* (Figs. 3 and 5). A major cluster of highly enriched sites is located in the sector surrounding the *segAB* locus, which points to a key functional role for this region in chromosome segregation. Most of the SegB sites are intragenic (Supplementary Fig. 9b, e), suggesting that SegB is involved in global chromosome organisation rather than transcriptional regulation. The chromosome of Sulfolobales exhibits discrete chromosomal interaction domains (CIDs)[10,46], analogous to those in bacteria[4]. The higher-order interactions of CIDs result in the establishment of two compartments, A and B[10,46]. The multiple *oriC* are located within the A compartment. ClsN, a non-canonical SMC protein, associates mostly with domains of the B compartment, thereby mediating its organisation[10,46]. Analogously, the high enrichment of SegB sites in the A compartment suggests that SegB may have an important role in the formation of this compartment specifically and, more broadly, in chromosome organisation in Sulfolobales. The short- and long-range activities on the DNA uncovered for SegB in vitro (Figs. 8 and 9) support the potential ability to mediate local domain formation and long-distance loci bridging via loops in the cell. It is tempting to speculate that the SegB binding to lower-affinity sites may mediate short-range interactions that contribute to the formation of chromosomal domains, whereas high-affinity sites may be anchoring points

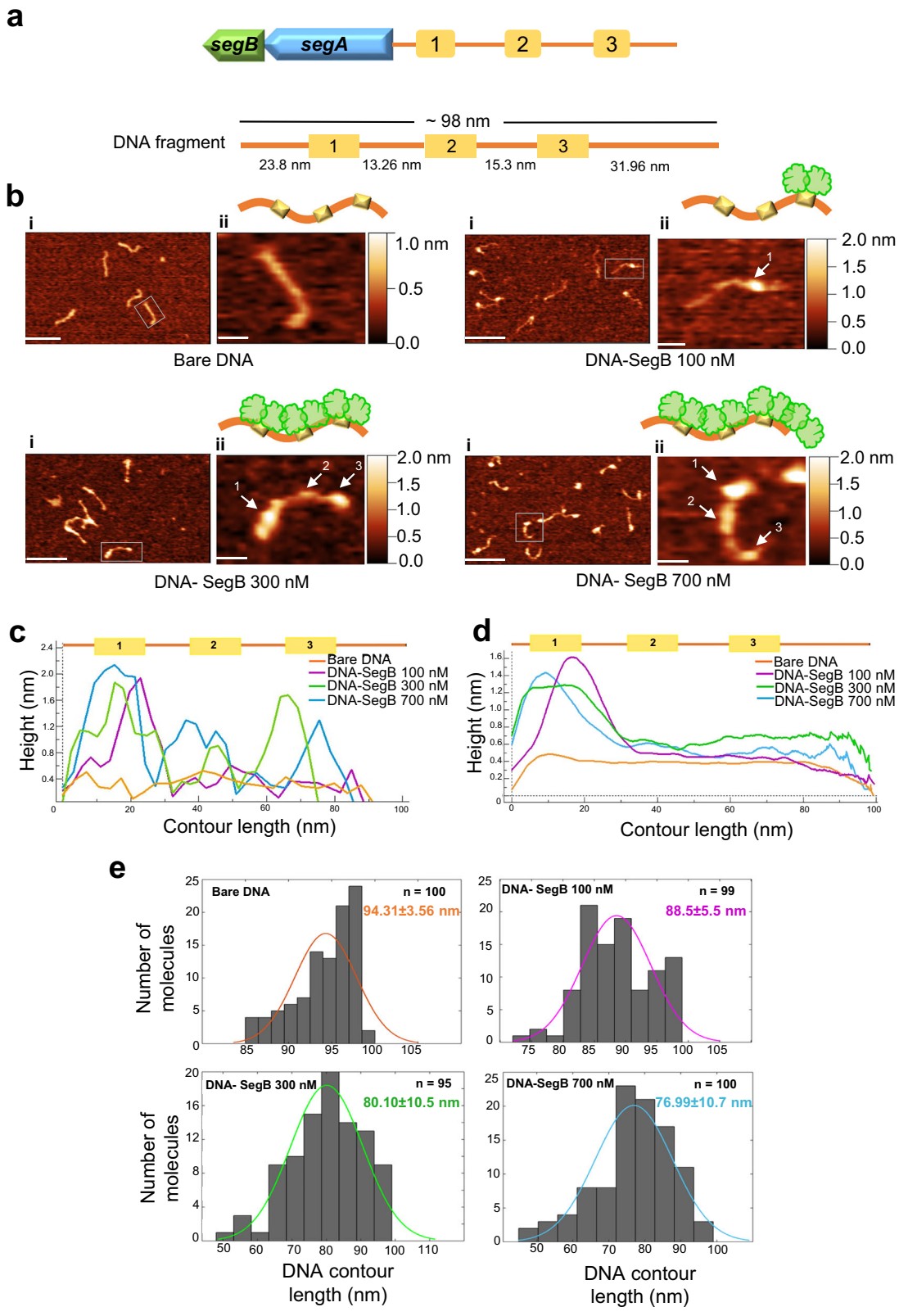

that bridge distantly located regions through long-range interactions. The synergy of site-specific DNA-binding by SegB and non-specific DNA-binding by SegA, in addition to the high-affinity SegA-SegB interaction, may confer a three-dimensional organisation to the chromosome that is instrumental for segregation. In the absence of such organisation, chromosome segregation is defective. Interestingly, the presence of SegB sites in genes encoding for reverse gyrase, RNA

polymerase B" subunit and coalescin in *S. solfataricus* and in the gene encoding the nucleoid-associated Alba protein in *S. acidocaldarius* points to a potential functional linkage to factors and processes involved in chromosome topology and compartmentalization, thereby reinforcing a role for SegB in chromosome organisation.

The bacterial chromosome segregation protein ParB binds as a dimer to *parS* sites. Subsequently, the bridging of dimers results in the

**Fig. 8 | SegB exhibits short-range DNA compaction activity. a** Schematic of the *segAB* locus and upstream sites (1, 2, and 3) bound by SegB (top); schematic of the 289 bp DNA fragment with details of the three binding sites used for AFM experiments. **b** Representative AFM images of bare DNA fragment (289 bp) or SegB-DNA complexes formed upon adding 100, 300 and 700 nM SegB. Scale in panel (i) = 100 nm and in panel (ii) = 20 nm. Arrows point at the DNA sites that are occupied by SegB. The sites labelled 1, 2 and 3 on the images correspond to those shown in panel a. The bars on the right side of the images provide a colour gradient reference for the height of particles. **c** Schematic of the 289 bp DNA fragment with the three SegB sites (not to scale) (top); plot showing height traces of the individual single particle of bare DNA (in panel b ii, top left), of DNA-SegB (100 nM) (in panel b ii, top right), of DNA-SegB (300 nM) (in panel b ii, bottom left), and of DNA-SegB (700 nM) (in panel b ii, bottom right) (bottom). **d** Schematic of the 289 bp DNA fragment showing the three SegB binding sites (not to scale) (top); average height traces of bare DNA molecules ($n = 100$), DNA-SegB (100 nM) particles ($n = 99$), DNA-SegB (300 nM) particles ($n = 95$) and DNA-SegB (700 nM) particles ($n = 100$). **e** Histograms of the contour length of bare DNA molecules and the DNA in the DNA-SegB complexes at different SegB concentrations. The mean of the contour length is indicated in each plot. The coloured line represents nonlinear Gaussian fit. Three independent biological replicates were performed for each sample with similar results. For the graphs source data are provided as Source Data file.

formation of a higher-order ParB-DNA condensate that mediates DNA compaction by recruiting adjacent chromosomal regions into the multimeric hub and by enabling the formation of loops connecting distant DNA regions[47–50]. Although SegB is evolutionarily unrelated to ParB, it is interesting that different domains of life deploy distinct DNA binding proteins to mediate chromosome compaction and segregation potentially using analogous principles and mechanisms.

The observation of multiple SegB foci on the nucleoid is consistent with the binding of the protein to multiple sites on the chromosome, and the formation of large patches underpins the potential of the protein to bridge distant domains in vivo (Fig. 7). Moreover, the widespread three-dimensional SegA meshwork through the nucleoid enables a web of SegA-SegB interactions that further contribute to chromosome organisation. The propensity of SegB foci to localise at the cell periphery, in cells with multiple foci, may signal the existence of potential tethers to the cell membrane (Supplementary Fig. 12). Anchoring loci to cellular landmarks facilitates chromosome spatial positioning and has been observed in both eukaryotic and bacterial cells[51–54]. Interestingly, very recent single-molecule localisation microscopy studies have shown that the nucleoid of rapidly growing *E. coli* cells assumes a membrane-proximal ring-like structure[55]. The mechanism underpinning the clustering of SegB foci towards the cell periphery deserves further investigation.

The conservation and, at the same time, diversity of SegB orthologues across archaea offer further insights into the functions of these proteins. The repeats identified in SegB from the *Metallosphaera* genus and their predicted folding into a distinctive β-helix raise questions about the function of this domain. Dali and FoldSeek comparisons for the predicted SegB revealed structural similarities to diverse proteins harbouring a β-helix, most of which are membrane- or cell surface-associated, including bactofilin, although the Z-score for this protein is modest (3.9). Nevertheless, this potential structural similarity may be particularly relevant, as bactofilins are bacterial cytoskeletal proteins that associate with the membrane[56], and localize the ParAB*S* complex to the subpolar regions of *Myxococcus xanthus*[57,58]. Although based on a predicted structure, these potential features of larger *Metallosphaera* SegB proteins may point again to a possible cell membrane association.

We propose a speculative model that integrates our findings with other data in the literature (Fig. 10). As previously reported, the chromosome of Sulfolobales is organised into two spatially segregated compartments A and B[10], each of which contains multiple CIDs[46]. SegA binds DNA non-site-specifically, forming a three-dimensional web of interactions and bringing together non-adjacent sites throughout the nucleoid, in both A and B compartments. SegB associates to multiple cognate DNA sites scattered across the chromosome, but mostly located in the A compartment. SegB binding results in short-range DNA compaction by bridging neighbouring sites within and across CIDs, and long-range DNA condensation by connecting long-distance sites across chromosomal domains. The biased localisation of SegB foci towards the cell periphery evokes the existence of potential tethers to the cell membrane

and the arc shape feature of the chromosome would ensure multiple anchoring points. The network of SegA-SegB high-affinity interactions further contributes to chromosome packaging. This SegAB-mediated genome organisation in turn, enables errorless chromosome segregation. ClsN acts primarily to organise the B compartment, and whether a crosstalk with SegAB exists remains to be investigated. Collectively, our results shed new light on significant facets of the mechanisms underpinning the function of the SegAB system in archaea and provide the foundations for a speculative model that points to SegB as an important player in coupling chromosome organisation and segregation.

## Methods

### Strains and growth media

*E. coli* DH5α [F⁻ *endA1 hsdR17* (r$_K^-$ m$_K^+$) *supE44 thi-1 recA1 gyrA96* (Nal$^r$) *relA1 deoR* Φ80*lacZ*ΔM15 Δ(*lacZYA-argF*) U169][59] was used for plasmid construction and propagation. Protein overproduction was performed in *E. coli* BL21 (DE3) CodonPlus-RIPL [F⁻ *ompT hsdS*(r$_B^-$ m$_B^-$) *dcm*+ Tet$^r$ *gal* λ(DE3) *endA* Hte [*argU proL* Cam$^r$] [*argU ileY leuW* Strep/Spec$^r$]. Strains were grown in Luria-Bertani (LB) broth or agar medium at 37 °C. Appropriate antibiotics for plasmid selection were included (ampicillin, 100 μg/mL; chloramphenicol, 30 μg/mL). *Sulfolobus acidocaldarius* MW001 [DSM639 Δ*pyrE* (*Saci*1597; Δ91–412 bp)][60] and *Saccharolobus solfataricus* PBL2025 [98/2 strain Δ(*sso3004-3050*)][61] were used for the construction of gene deletion strains. The *S. solfataricus* PBL2025 strain presents a 58 kbp deletion in the chromosome that includes the *lacS* gene. This selectable marker replaced the *segB* gene in Δ*segB*. *S. solfataricus* PBL2025 and P2[62], as well as *S. acidocaldarius* MW001, were used for the ChIP-seq experiments. Cultures were grown at 75 °C in Brock's medium[63] supplemented with either 0.1% tryptone or NZ-amine, 0.2% sucrose and uracil (20 μg/mL), when necessary. *S. solfataricus* strains that carry the simvastatin-resistance selection marker were grown in Brock's medium supplemented with simvastatin (15-20 μM).

### Plasmids

All plasmids used in this study are listed in Supplementary Table 4 (Supplementary Information file). The pET-*sso35* plasmid[11] was used for *S. solfataricus* SegB overproduction. The *S. solfataricus segA* gene including stop codon was cloned into pET22b (Novagen) using *Nde*I and *Xho*I restriction sites. The *S. acidocaldarius segB* gene was cloned into pET-22b deploying *Nde*I and *Xho*I restriction sites. Plasmid pET2268[64] was used to construct the *S. solfataricus* Δ*segB* strain and pSVA406[60] was used as a template for constructing *S. acidocaldarius* Δ*segB* and Δ*segAB* strains. For complementation studies, the *segB* gene was amplified from *S. solfataricus* P2 genome by PCR and was then cloned into the pSSR vector[65], which carries a simvastatin resistance marker *hmg* downstream of the arabinose promoter, using *Nde*I and *Cla*I restriction sites, generating p*segB*.

For AFM studies, plasmid pNER803 was constructed by cloning a 1028 bp DNA fragment amplified by PCR using *S. solfataricus* P2 genome into pUC18 digested with *Sph*I and *BamH*I. This fragment contains the SegB high-affinity binding site (5′-GTAGAGTCTAGACT-

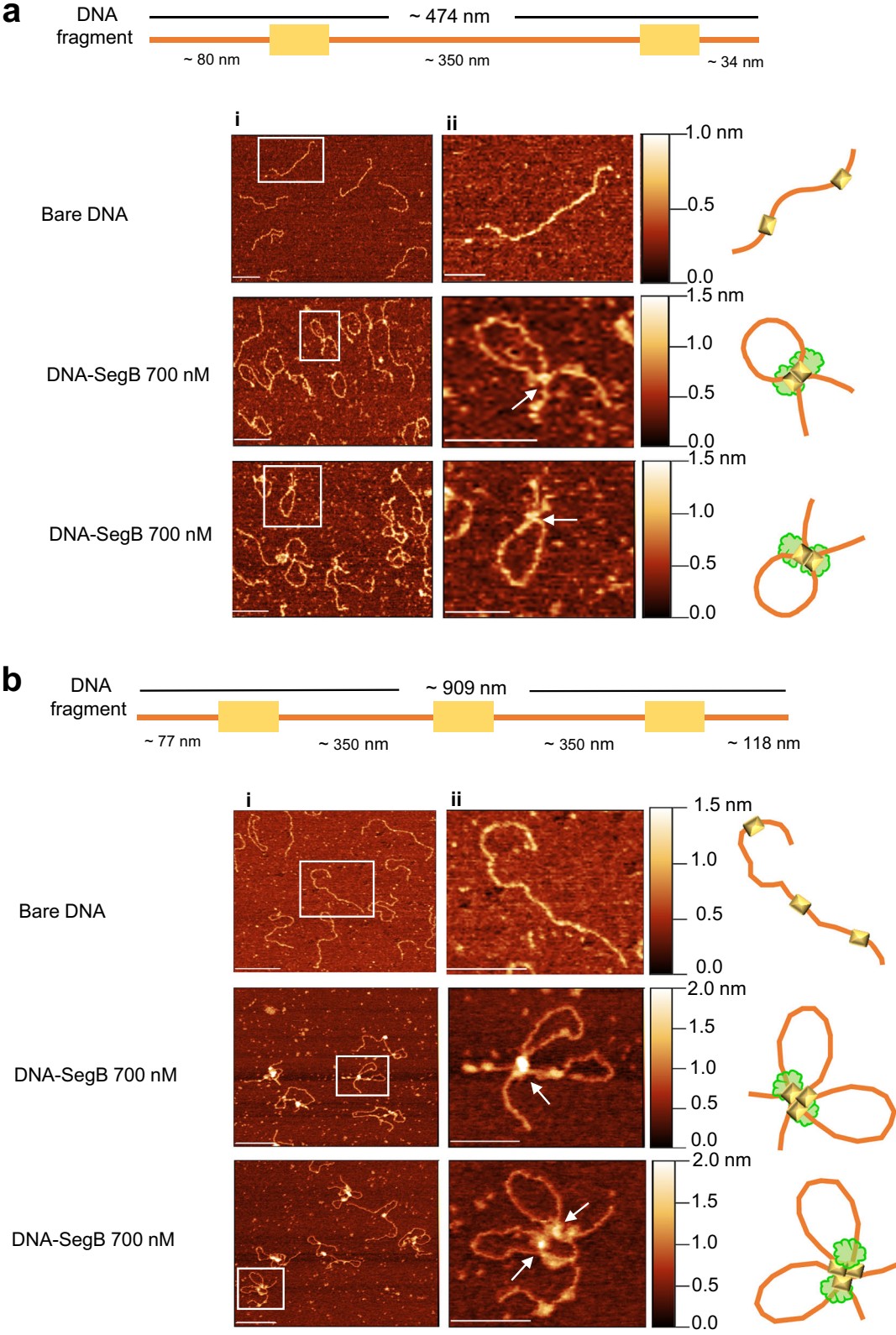

3'). A 126 bp DNA fragment that contains the same high affinity site was then amplified from *S. solfataricus* P2 genome and cloned into pNER803 digested with *BamH*I and *Kpn*I, generating pNER804. Another plasmid, pNER805, containing three high-affinity SegB binding sites, was constructed by cloning a 1028 bp DNA fragment amplified from the *S. solfataricus* P2 genome into pNER804 digested with *Sph*I and *Hind*III.

## Construction of *S. solfataricus* and *S. acidocaldarius* deletion strains

*S. solfataricus* Δ*segB* deletion strain was constructed by PCR amplification of the 823 bp sequence upstream and 1150 bp sequence downstream of the *segB* gene using the primers listed in Supplementary Table 3 (Supplementary Information file). The upstream region was cloned into the vector pET2268 upstream of the β-galactosidase-

**Fig. 9 | SegB bridges long-distance sites forming loops and loop hubs in vitro.** **a** (Top) Schematic of the DNA fragment (1.4 kbp) used for the experiments, which contains two identical high-affinity sites (5′ -GTAGAGTCTAGACT- 3′) at a distance of - 350 nm (1028 bp). (Bottom) Representative AFM images of the bare DNA fragment and SegB-DNA complexes (700 nM SegB). The ii panels are zoomed-in images. The white arrows point at the bridging point between the two SegB binding sites. The bars on the right side of the images provide a colour gradient reference for the height of particles. Scale bar in i panels = 200 nm and in ii panels = 100 nm. Three independent experiments were performed with similar results. **b** (Top) Schematic of the DNA fragment (2.7 kbp) used in the experiments, which contains three identical high-affinity sites (5′-GTA-GAGTCTAGACT-3′) at a distance of -350 nm (1028 bp) from one another. (Bottom) Representative AFM images of the bare DNA fragment and SegB-DNA complexes (700 nM SegB). The ii panels are zoomed-in images. The arrows point at the bridging point between the two SegB binding sites. The bars on the right side of the images provide a colour gradient reference for the height of particles. Scale bar in i panels = 200 nm and in ii panels = 100 nm. Three independent experiments were performed with similar results.

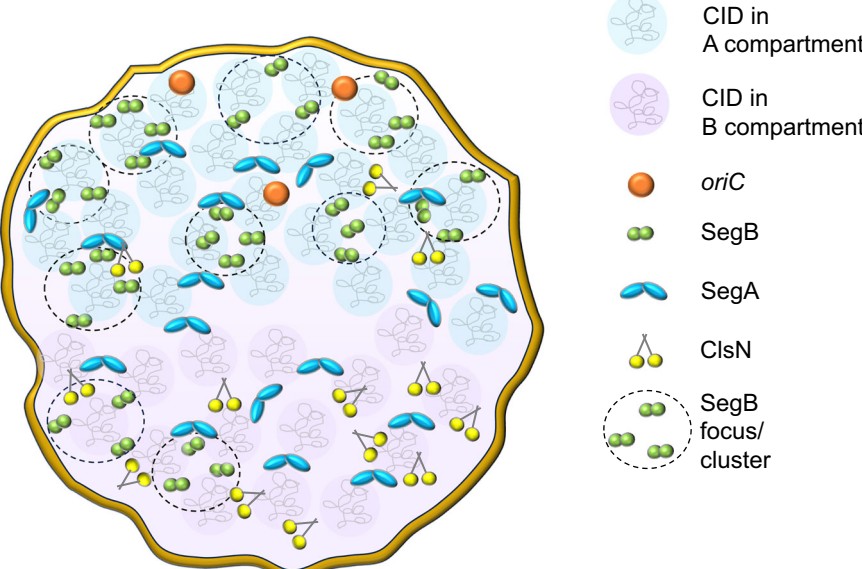

**Fig. 10 | I Model summarizing the results and illustrating the role of SegAB in chromosome compaction and organisation.** The chromosome is organised into two compartments[10], A coloured in blue and B depicted in lilac. Each compartment contains multiple CIDs[46]. SegA (blue dimers) binds DNA non-site-specifically, forming a 3-dimensional web of interactions and bringing together non-adjacent sites throughout the nucleoid (A and B compartments). SegB (green dimers) recognises and binds to multiple cognate sites scattered across the chromosome, although predominantly located in the A compartment. SegB binding induces short-range DNA compaction upon associating with neighbouring sites within and across CIDs, and long-range DNA condensation by interacting with long-distance sites across chromosomal domains. The high-affinity interactions between SegA and SegB add an extra layer of interconnectivity that contributes further to chromosome packaging and folding. This level of genome organisation in turn enables accurate chromosome segregation. ClsN acts primarily in the B compartment, and whether a crosstalk with SegAB is at play remains to be investigated. SegB forms multiple foci or clusters on the nucleoid that are able to coalesce into large patches. The biased localisation of SegB foci towards the cell periphery, in cells with multiple foci, evokes the existence of potential tethers to the cell membrane or unknown interacting factors.

encoding gene (*lacS*), using *Kpn*I and *Nco*I restriction sites. The downstream region was cloned into pET2268 downstream of *lacS*, using *Bam*HI and *Not*I restriction sites. The upstream and downstream regions and the intervening *lacS* were amplified by PCR using external primers to generate a DNA fragment, which was then *Dpn*I treated and used to transform *S. solfataricus* PBL2025 cells for homologous recombination by a double crossing-over event. Transformants were selected by growth in lactose-containing Brock's medium over 5 days. These were then transferred to fresh lactose-containing medium and grown for 13 days. The cultures that grew were streaked onto solid gelrite Brock's medium plates containing 0.2% tryptone, no sugar, pH 3.5, and incubated at 76 °C for 5 days. The presence of *lacS* as confirmation of the double crossover event was tested by blue-white screening by spraying plates with X-gal (5 mg/mL). Transformants were screened by colony PCR to test for the absence of the *segB* gene.

Construction of the *S. acidocaldarius* deletion mutants Δ*segB* and Δ*segAB* involved PCR amplification of *S. solfataricus pyrBEF* marker using oligonucleotides which also contained sequences that anneal to the ~50 bp of sequence upstream and ~50 bp downstream of the genes to be deleted. The purified PCR products (300 ng) were used to transform *S. acidocaldarius* MW001 competent cells that were plated onto gelrite Brock's medium supplemented with 0.1% NZ-amine and 0.2% sucrose, but lacking uracil to select for the *pyrBEF* marker. Transformants grown on the plates were screened by colony PCR. Gene deletions were confirmed by Sanger sequencing of PCR products and full genome sequencing.

## Chromatin immunoprecipitation (ChIP) and Illumina sequencing in *S. solfataricus*

*S. solfataricus* P2, PBL2025 and Δ*segB* strains were grown at 75 °C in 200 mL Brock's medium, pH 3.0, supplemented with 0.1% tryptone. When the cells reached mid-exponential growth phase (OD$_{600}$ 0.4 – 0.6), the cultures were removed from 75 °C and cooled down at room temperature, before cross-linking with 1% formaldehyde, shaking at 37 °C for 30 min. The cross-linked reactions were stopped by adding 125 mM glycine and incubated at 37 °C, with shaking for 15 min. Cross-linked cells were harvested by centrifugation at 3220 × g for 10 min[66]. Cells were washed three times with 1x PBS and resuspended in 1 mL of Buffer 1 (20 mM HEPES pH 7.9, 50 mM KCL, 10% glycerol plus EDTA-free protease inhibitors). Cells were then sonicated on ice (at power 2, 3 s ON and 9 s OFF with a total operation time of 9 min) to shear the DNA[66]. The majority of the sheared DNA fragments ranged between 100–500 bp. The cells were then centrifuged for 20 min (15115 × g, 4 °C) and the supernatant was transferred into a clean

microfuge tube. Fifty microliters of the supernatant were transferred into a separate microfuge tube as a control (INPUT sample) after adjusting the buffer conditions (to 10 mM Tris–HCl pH 8.0, 150 mM NaCl and 0.1% NP-40) and stored at −20 °C[67].

The rest of the cross-linked, sonicated and centrifuged sample was incubated overnight at 4 °C with polyclonal anti-SegB rabbit antibodies. The next day, the sample containing the antibody-protein-DNA complex was mixed with 50 μL of Pierce™ Protein A Magnetic Beads (Thermo Fisher), which were previously washed with IPP150 buffer (10 mM Tris–HCl, pH 8.0, 150 mM NaCl and 0.1% NP-40) and incubated for 4 h at 4 °C. The beads were washed five times with 1 mL IPP150 buffer for 2 min at 4 °C and then washed twice with 1x TE buffer (10 mM Tris–HCl, pH 8.0, 1 mM EDTA).

DNA-protein complexes were eluted from Pierce™ Protein A Magnetic Beads by incubating the mixture with 150 μL of Elution Buffer (50 mM Tris–HCl, pH 8.0, 10 mM EDTA, 1% SDS) for 15 min at 65 °C, then with 100 μl of TE buffer supplemented with 1% SDS for further 15 min at 65 °C. The supernatant (ChIP sample) was transferred into a new microfuge tube. Two hundred microliters of TE buffer containing 1% SDS were added to the INPUT sample. ChIP and INPUT samples were incubated overnight at 65 °C to reverse the cross-linking. Subsequently, the DNA was purified from ChIP and INPUT samples using a PCR purification kit (Qiagen) and eluted in 40 μl 0.1 x EB buffer.

The purified DNA was then used for qPCR and to construct ChIP-Seq libraries using NEBNext® Ultra™ II DNA Library Prep Kit for Illumina® (New England Biolabs). Libraries were sequenced using Illumina Miseq system (paired end 75 bp reads) or Illumina Hiseq 3000 (paired end 150 bp reads) at the University of Leeds Next Generation Sequencing Facility. The ChIP-seq profiles were generated using customised R scripts where the X-axis represents the genomic position in base pair and the Y-axis represents the number of reads per base pair per million mapped reads (RPBPM). Fifteen high-enrichment and twenty-four low-enrichment peaks were identified, the latter defined as peaks with a RPBPM value inferior to 5. ChIP–seq data were deposited in the Gene Expression Omnibus.

SegB DNA-binding motifs present in ChIP-seq peaks were identified using MEME-ChIP[68]. FIMO[69] was deployed to identify analogous SegB motifs on the chromosome of other archaeal genera, using the *S. solfataricus* SegB DNA-binding consensus sequence as a query.

### Chromatin Immunoprecipitation and Illumina sequencing in *S. acidocaldarius*

*S. acidocaldarius* MW001 and ΔsegB were grown in 200 mL Brock's medium, pH 3.0, containing 0.1% NZ-amine, 0.2% sucrose and uracil (20 μg/mL) (only for MW001) at 75 °C until the exponential phase ($OD_{600}$ 0.4-0.5). Cultures were cooled at room temperature for 10-15 min before cross-linking with 1% formaldehyde at 37 °C for 30 min in a shaker incubator. Cross-linking reactions were quenched with glycine (250 mM) and incubated for 15 min at 37 °C with shaking. Cells were harvested by centrifugation at 9302 g, 10 min, 4 °C. Pellets were washed three times with 1X PBS. Excess PBS was removed by inverting the tubes on a tissue for ~5 min. Cross-linked cell pellets were then stored at -80 °C overnight (or up to a week).

Cross-linked pellets were thawed on ice for 10-15 min and resuspended in 1 mL of Buffer 1 + protease inhibitors (20 mM Hepes-KOH, 50 mM KCl, 10% glycerol and protease inhibitor cocktail) by pipetting gently. Cells were then transferred to 1.5 mL microcentrifuge tubes and sonicated using Soniprep 150 probe-type sonicator (15 ON, 15 OFF, 13 cycles, 8 Amp). Upon sonication, cells were centrifuged at max speed for 20 min at 4 °C. Supernatant was transferred into a fresh 1.5 mL tube and the following were individually added to each sample: 10 mM Tris-Cl, pH 8.0, 150 mM NaCl, 0.1% NP40. The samples were mixed by inverting the tubes and then were centrifuged at 15115 g for 2 min at 4 °C. Supernatant was transferred to a fresh 2 mL tube on ice. Fifty microliters from each sample were aliquoted in another 2 mL tube as

input fraction (control) and stored at -20 °C overnight. Meanwhile, overnight swollen Protein A beads were rotated at 4 °C on an orbital rotator for 30-40 min and washed twice with IPP150 Buffer (10 mM Tris-HCl, pH 8.0, 150 mM NaCl, 0.1% NP40) by centrifugation at 8944 g for 30 s.

Fifty microliters of the rotated beads were added to the above supernatant for at least 1 h in the cold room to allow pre-cleaning of the supernatant to reduce non-specific binding. Supernatant was collected in a fresh 2 mL tube by centrifugation at 15115 g for 10 min at 4 °C. Fifty microliters of rabbit polyclonal anti-segB antibody (5.7 mg/mL) were added to all samples, that were then rotated overnight at 4 °C. Two hundred μL of washed Protein A beads (stored at 4 °C from the previous day) were added to each supernatant + antibody sample, which were rotated at 4 °C for at least 4 h and then centrifuged at 8944 g for 30 s at 4 °C. The supernatant was discarded, and the beads were washed five times with IPP150 Buffer and then twice with 1X TE Buffer. One hundred and fifty microliters of Elution Buffer (50 mM Tris-Cl pH 8, 10 mM EDTA, 1% SDS) were added to the washed beads and mixed by vortexing. Samples were incubated at 65 °C for 15 min and centrifuged at 15115 g for 5 min at room temperature. The supernatant was transferred to a fresh 2 mL tube. DNA extraction was repeated using 100 μL of 1X TE buffer containing 1% SDS, by vortexing the beads and incubating at 65 °C for 5 min. Supernatant (the ChIP fraction) was collected, pooled with the first one, and incubated at 65 °C overnight.

Input samples stored at -20 °C on the previous night were thawed and incubated at 65 °C with 200 μL of TE buffer containing 1% SDS to reverse the cross-linking overnight. DNA from the ChIP and input fractions was purified using a PCR purification kit (Qiagen), according to the manufacturer's protocol. DNA was eluted in 40 μL of sterile ddH2O. The purified DNA was then used to construct libraries suitable for Illumina sequencing using the NEXT Ultra II library preparation kit (New England Biolabs). ChIP libraries were sequenced on the Illumina Miseq at the Tufts University Genomics facility.

For analysis of ChIP-seq data, Miseq Illumina short reads (50 bp) were mapped back to the *S. acidocaldarius* reference genome (NCBI NC_007181.1) using Bowtie2 VN:2.5.3[70,71] according to these run conditions: -D 15 -R 2 -N 1 -L 22 -i S,1,1.15. ChIP-seq peaks were called manually using Artemis[72] and potential SegB binding sites were identified using MEME-ChIP[68]. Binding sites were mapped onto the *S. acidocaldarius* genome and analysed for their occurrence at intra/intergenic regions along with the compartment (A or B) in which the peaks were located. ChIP-seq profiles were plotted with the X-axis representing genomic positions and the Y-axis is the RPBPM using custom R scripts.

### Fluorescence microscopy

To assess nucleoid presence and morphology, *S. solfataricus* and *S. acidocaldarius* strains were grown in Brock's medium containing 1% NZ-amine (and uracil if required) to an $OD_{600}$ ~ 0.4-0.5 and stained with DAPI according to the protocol previously described[11]. Images were collected using a Zeiss LSM 710 inverted confocal microscope and analyzed with Volocity (v5.5, Perkin Elmer). Plots and statistical analyses were performed using GraphPad Prism v. 10.3.0.

### Immunofluorescence microscopy and image analysis

Early exponential *S. solfataricus* cells were harvested and fixed in either 2.5% paraformaldehyde or 70% ethanol. The cell membrane was permeabilised using buffer (50 mM glucose, 20 mM Tris-HCl, pH 7.5, 10 mM EDTA, and 0.2% Tween 20) and incubated at 25 °C for 7 min. Cells were washed three times with phosphate-buffered saline (PBS) to remove all traces of detergent. Ten microliters of cell suspension were spotted on poly-L-lysine-coated coverslips and air-dried. The coverslips were then rehydrated with PBS, and blocked with 2% bovine serum albumin (BSA, Sigma) in PBS for 15 min at room temperature. Cells were incubated overnight at 4 °C with purified polyclonal anti-

SegA, anti-SegB primary antibodies diluted in 2% BSA, 1X PBS, and 0.05% Tween 20. The anti-SegA antibody was used at a dilution of 1:100, the anti-SegB antibody was used at a dilution of 1:250. Cells were washed with PBS and then incubated in the dark at room temperature for 1 h with goat Alexa Fluor 555-conjugated anti-rabbit IgG (Invitrogen, catalogue # A-21428) diluted in 2% BSA, 1X PBS, and 0.05% Tween 20. The secondary antibodies were used at a dilution of 3:1000, and cells were washed with PBS. Ten microliters of DAPI (Sigma-Aldrich) and a drop of *SlowFade*® Gold Antifade (Thermo Fisher) were spotted on the slide. The coverslip was then placed over DAPI. The edges of the coverslip were sealed with nail polish.

Imaging was performed on a Nikon Eclipse Ti-E inverted microscope equipped with Plan Apo 100x Ph oil (N.A. 1.45) and Plan Apo VC 100x oil (N.A. 1.4) objective lenses coupled to an Andor Zyla 5.5 sCMOS camera. Images were processed using Fiji where brightness and contrast features were utilised for optimal observation of cell morphology, protein and DNA visualisation. Processed and saturated images were analyzed using a customised MATLAB script. Graphs were generated using either GraphPad Prism or SigmaPlot.

## Flow cytometry

*S. solfataricus* and *S. acidocaldarius* cells were grown at 75 °C in Brock's medium, pH 3.5, (50 mL) plus supplements. One mL aliquots were collected from exponentially growing cultures (OD$_{600}$ ~ 0.2). Cells were fixed with 70% ice-cold ethanol for a minimum of 30 min on ice and stored at 4 °C until use. Fixed cells were harvested by centrifugation (8 min at 6000 g at 4 °C) and washed once by resuspension in 1 mL of buffer (10 mM Tris-HCL, pH 7.5, 10 mM MgCl$_2$). Cells were resuspended in 500 μL buffer and mixed with a freshly made 200-fold dilution of Quant-iT™PicoGreen® fluorescent reagent (Invitrogen™) to stain the DNA. Samples were incubated in the dark for 30 min. Flow cytometry experiments were performed using the CytoFLEX LX (Beckman Coulter) by exciting the PicoGreen-stained cells with the 488 nm laser. Emitted photons were detected by avalanche photodiode detector after passing through reflective bandpass filter 530/30 nm. 1,000,000 events per sample were recorded at a flow rate of 30 μL /minute. Either two or three biological replicates were analysed. Data were analysed using FlowJo 10.10.0 software.

## Total RNA isolation and Reverse Transcription into cDNA

Total RNA was extracted from exponentially growing *S. solfataricus* P2 cultures using RNeasy mini kit (Qiagen). Total RNA was treated with recombinant DNase I (Thermo Fisher) according to the manufacturer's protocol to remove any residual genomic DNA. One microgram of total RNA was used to synthesise the first-strand cDNA using SuperScript IV reverse transcriptase (Invitrogen) and gene-specific primers AK51 and AK52 (Supplementary Information file). One microgram of cDNA was used to perform PCRs using GoTaq polymerase (Promega) and primers annealing to the 5′-end and 3′-end of the *segAB* operon. PCR products were resolved on a 1% agarose gel and visualised using SYBR Safe.

## Homology searches, gene neighbourhood conservation and phylogenetic analyses

The webFlaGs programme[17] was deployed to identify SegB orthologues and to investigate the genomic context of the *segB* gene in different archaeal genera. After inputting *S. solfataricus* SegB sequence, the software performed a BLASTP search against the microbial RefSeq database to identify orthologues using an E value cut-off of 1e⁻3. The FlaGs programme clustered proteins encoded by flanking genes based on homology, using Hidden Markov Model-based method Jackhmmer that is part of the HMMER package[73]. The clustering was performed using an E value cut-off of 1e⁻10, three Jackhmmer iterations and a maximum of four upstream and downstream genes. FlaGs generated a phylogenetic tree of SegB orthologues by using the tree-building feature of ETE v3 Python environment[74].

## Structure prediction and AlphaFold 2 and 3 model analysis

The sequences of *M. hakonensis* and *M. sedula* SegB orthologues were used to generate a predicted model of the structures through AlphaFold 2[20,22] running ColabFold[75] from within ChimeraX[76]. The PDB files of the predicted structures are provided as Supplementary Data (Supplementary Data 1, 2, 3 and 4). The PDB of the best models was used to search for structural homologues with the Dali server[24] and FoldSeek[25]. For AlphaFold 3 structure predictions, the server was accessed on 3/05/2025. The best model has been included as Supplementary Data 5. The superimposition of SufB and the AF2-predicted structure of *M. hakonensis* monomeric SegB was obtained through FoldSeek and the PDB opened in ChimeraX v.1.9.

## Statistical analyses

Statistical analyses were performed using GraphPad Prism (version 10.4). Data were compared using the two-tailed Wilcoxon test and the one-tailed Kruskal-Wallis test followed by the Dunnett's multiple comparison test. P and n values were reported in the figure legends.

## Protein overproduction and purification

Proteins were overproduced in *E. coli* BL21 (DE3) Codon Plus cells. Cultures were grown to an OD$_{600}$ of 0.4-0.6 at 37 °C, shaking at 120 rpm and then induced with IPTG (1 mM final concentration). Proteins were overproduced for three hours at either 37 °C (SegB) or 30 °C (SegA). For His-tagged proteins the cell pellets were resuspended in binding buffer (20 mM HEPES-KOH pH 7.5, 1 M NaCl, 5 mM imidazole) supplemented with 0.5 mM phenylmethylsulfonyl fluoride (PMSF) and one tablet of complete protease inhibitor (Roche). Cells were lysed using lysozyme (0.2 mg/mL), incubating at 30 °C for 30 min before sonicating for 30 s pulses, repeating 7 times. Lysed cells were then heated to 76 °C for 15 min to denature native *E. coli* proteins. The supernatant was clarified by centrifugation at 15,000 g for 30 min at 4 °C.

The His-tagged proteins were purified by Ni²+ affinity chromatography and eluted with buffer (20 mM HEPES-KOH pH 7.5, 1 M NaCl, 1.5 M imidazole). The protein was then exchanged into 20 mM MES, pH 6.0, using a Hi-Trap desalting column (Cytiva) and loaded onto a MonoS column (Cytiva). Protein was eluted using elution buffer (20 mM MES pH 6.0, 1.5 M NaCl). The final fractions were buffer-exchanged into storage buffer (20 mM HEPES-KOH pH 7.5, 50 mM NaCl, 10% glycerol) by using a Hi-Trap desalting column (Cytiva). To purify native SegA, the cell pellets were resuspended in 20 mM HEPES, pH 7.5, supplemented with 0.5 mM PMSF and one tablet of complete protease inhibitor (Roche) and lysed as above. The clarified cell lysate was loaded onto a heparin sepharose 6 FF resin column (Cytiva) and eluted with 1.5 M NaCl. Protein was buffer exchanged and a second chromatography step using MonoS was carried out as for His-tagged proteins.

## Electrophoretic mobility shift assay (EMSA)

Band-shift assays were carried out using different biotinylated DNA fragments generated by PCR amplification. SegB (0-1000 nM) was incubated with the DNA fragment (1-5 nM) for 20 min at 37 °C in binding buffer (10 mM Tris-HCl pH 7.5, 50 mM KCl, 1 mM DTT, 5 mM MgCl$_2$, 2.5% glycerol and 0.05% NP-40), containing 1 μg of competitor poly(dIdC) DNA. The reactions were run on 0.8-1.5% agarose or 5% acrylamide gels in 0.5 x TBE at 100 V. The DNA fragments were then electroblotted onto a positively charged nylon membrane and detected by using the LightShift chemiluminescence kit (Thermo Fisher).

## DNase I footprint

Reactions were assembled as for EMSAs in a 20 μL reaction volume and DNase I footprint was performed as previously described[11].

## Microscale thermophoresis (MST)

A SegB aliquot (5-20 µM) was labelled with the amine-reactive red dye NT-647 (NanoTemper Technologies). The fluorescence of the labelled protein was measured to determine the amount to be included in the MST reactions. Ideally, 200 fluorescence units were required. The unlabelled protein was serially diluted 1:1 using MST buffer (10 mM Tris-HCl, pH 7.4, 100 mM NaCl, 0.05% Tween 20) over fifteen tubes. Each dilution was mixed with 10 µL of labelled SegB protein. The reactions were transferred into hydrophilic glass capillaries (Nano-Temper Technologies). Labelled SegB was used undiluted at a final concentration of 304 nM. MST measurements were carried out using the following parameters: 20% LED power, 40% MST power, 5 s delay before heating, 30 s during MST, 5 s delay after heating, 15 s delay. Binding data were analysed using NTanalysis software (NanoTemper Technologies).

## Atomic force microscopy (AFM)

A 289 bp DNA fragment was amplified by PCR using *S. solfataricus* P2 chromosomal DNA as template with forward primer 5′-ACGCTATG-TAGGCTGAAGCTA-3′ and reverse primer 5′-AGTAAGATAAATTTACT GAAGTCG-3′. To generate a DNA fragment containing two identical SegB high-affinity binding sites separated by 1028 bp, a DNA fragment containing the sequence 5′-GTAGAGTCTAGACT-3′ (peak D motif) was amplified from *S. solfataricus* P2 chromosome using forward primer 5′-ATAATAGCATGCAAGGGAAGTAGAG-3′ and reverse primer 5′-ATAA-TAGGATCCAGACTGGCTAAGAC-3′. This fragment was cloned into pUC18 digested with *Sph*I and *Bam*HI, generating pNER803. A 126 bp fragment containing the peak D motif was amplified from *S. solfataricus* P2 chromosome using forward primer 5′- ATAATAGGATCCAAG GGAAGTAGAG-3′ and reverse primer 5′-ATAATAGGT ACCTAATTAA-TAGTTAACAC-3′ and cloned into pNER803 digested with *Bam*HI and *Kpn*I, generating pNER804. This plasmid was digested with *Nde*I and *Kpn*I producing a 1388 bp DNA fragment.

To generate a DNA fragment containing three identical SegB high-affinity binding sites (3 peak D motifs) separated by ~1 kbp, a DNA fragment containing the motif was amplified from *S. solfataricus* P2 chromosome using forward primer 5′- ATAATAAAGCTTAAGGAAGTAG AGTC-3′ and reverse primer 5′- ATAATAGCATGCAGACTGGCTAAGAC-3′. This fragment was then cloned into pNER804 generating pNER805. This plasmid was digested with *Nde*I and *Sap*I to produce a 2672 bp fragment.

Purified recombinant proteins were used at different concentrations by diluting the protein in storage buffer (50 mM HEPES-KOH pH 7.0, 50 mM KCl, 10% glycerol). DNA (1 ng/µL) was mixed with the protein in a total volume of 20 µL. An aliquot of DNA (5 nM) was mixed with different protein concentrations (0-700 nM) to achieve different protein:DNA ratios. The reaction buffer contained 10 mM Tris-HCl pH 7.5, 10 mM KCl, 2.5% glycerol and 10 mM $MgCl_2$. The reaction mixture was incubated for 20 min at 37 °C, allowed to cool down for 1 min and then deposited on a freshly cleaved mica surface for 5 min. The mica was then rinsed with 1 mL filtered Milli-Q water and dried by a gentle stream of filtered air.

Images were acquired with a Bruker BioScope Resolve AFM microscope in air at room temperature using PeakForce quantitative nanomechanical (QNM) mapping in air mode. Super sharp silicon nitride tips were purchased from Bruker (SCANASYST-AIR-HR, SAA-HPI-SS). Images were processed (or flattened and levelled) using Gwyddion (http://gwyddion.net/) or WSxM software[77]. Bare DNA and DNA-protein contour lengths were manually traced and measured using WSxM software. The data were binned into histogram using MATLAB or GraphPad Prism (version 8.0) and were fitted to a Gaussian distribution. The mean of the DNA length and standard deviation were extracted from the Gaussian data. Bare DNA and DNA-protein height profiles were generated using WSxM software[77].

## Material availability

Requests for biological materials should be addressed to Daniela Barillà.

## Reporting summary

Further information on research design is available in the Nature Portfolio Reporting Summary linked to this article.

## Data availability

All data generated in this work are included in the manuscript and supplementary files. ChIP-seq data are available at GEO: project GSE169604 [https://www.ncbi.nlm.nih.gov/geo/query/acc.cgi?acc=GSE169604] and GSE297822 [https://www.ncbi.nlm.nih.gov/geo/query.cgi?acc=GSE297822]. Source data are provided with this paper.

## Code availability

The custom R code used for plotting ChIP-seq profiles can be found at https://github.com/TungLeLab/DSM639_Sacidocaldarius_ChIP_seq[78]. The MATLAB code used to analyze immunofluorescence microscopy images can be found at https://github.com/BarillaLab/Immunofluorescence_Analysis[79]. The codes can be accessed without restrictions.

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

## Acknowledgements

This work was supported by grants from The Leverhulme Trust [RPG-245] and the Biotechnology and Biological Sciences Research Council [BB/M007839/1, BB/R006369/1] (to D.B.) and [BB/X00645X/1] (to D.B. and T.B.K.L.); from the Royal Society Fellowship Renewal URF\R\201020, the Lister Institute fellowship and the BBSRC funded Institute Strategic Program Harnessing Biosynthesis for Sustainable Food and Health (HBio) (BB/X01097X/1) (to T.B.K.L.). We thank Finbarr Hayes for valuable comments on the manuscript, Sukhveer Mann in the Bioscience Technology Facility (University of York) for outstanding advice on flow cytometry, Andrew Leech in the Bioscience Technology Facility (University of York) for advice on protein interaction studies, Rosalyn Leaman and Christoph Baumann for assistance in developing codes for image analysis, Arthur Charles-Orszag and Dyche Mullins for openly sharing data before publication, and Qunxin She for the kind gift of the pSSR plasmid.

## Author contributions

A.F.K., I.W.N., N.R., P.P., T.B.K.L., S.V.A. and D.B. designed research. A.F.K., I.W.N., N.R., P.P., J.R. and N.T.T. performed research. J.R. and S.V.A. contributed new reagents/analytical tools. A.F.K., I.W.N., N.R., P.P., T.B.K.L. and D.B. analyzed the data. A.F.K., I.W.N., N.R. and D.B. wrote the paper with input from all other authors. D.B. and T.B.K.L. acquired funding.

## Competing interests

The authors declare no competing interests.
