## [Transparent Peer Review file · Nature Communications]

Coupling chromosome organization to genome segregation in Archaea

Corresponding Author: Professor Daniela Barillà

Version 0:

Reviewer comments:

Reviewer #1

(Remarks to the Author)

Review: Coupling Chromosome organization to genome segregation in Archaea

This is an interesting study and presents an impressive body of work that significantly advances our understanding of the chromosome segregation processes in Thermoproteota archaea, specifically the SegAB partition system. Crucially, the work provides insights into how these proteins organise and compact chromosomal DNA, seemingly in one of the two major spatial compartments within cells of the Sulfolobales and also highlights interesting putative associations of the compaction/segregation processes with cell division.

The study deploys a broad range of experimental approaches starting with bioinformatics and structural modelling to explore the distribution and likely structures of the SegAB division system. A C-terminal RHH domain in SegB is highlighted and the presence of this domain in other archaeal cell cycle regulator proteins is discussed.

In vivo deletion segB and SegAB is shown to cause chromosome segregation and cell division defects, with anucleate and cells displaying aberrant nucleoid morphology, consistent with chromosome segregation defects.

ChIP-seq approaches are then used in several Thermoproteota species to examine the binding localisation of these proteins across these archaeal genomes, and these studies reveal that SegB binds preferentially in the 'A' compartment of the cell that hold the replication origins prior to cell division, and typically the transcriptionally active components. Binding sites are then confirmed in vitro by band-shift and footprinting techniques that reveal lower and high affinity binding sites. The subcellular localisation of the SegA and SegB proteins are also demonstrated by immunofluorescence microscopy, while SegA to SegB binding is confirmed in vitro by microscale thermophoresis MST.

Exciting novel findings were also made towards the end of the study with the in vitro observation of localised DNA compaction following binding at the three Seg B recognition sites at the segAB by AFM. These data also provided further evidence of binding to higher and lower affinity sites. Finally, strikingly clear loop formation is demonstrated in vitro by AFM on longer DNA substrates by bridging long-distance sites, suggesting at least two novel DNA compaction modes via SegB in the Sulfolobales.

The article is for the large part well written, but the clarity could perhaps be slightly improved in places for a more generalist audience. For example, the A and B chromosome compartments probably require slightly more introduction. Also, while the abstract is clear enough in general it might be an idea to detail some of the experimental approaches that lead to the reported discoveries, and importantly there is reference to 'the' Short-range DNA compaction and long term looping – but there is no prior introduction to these events! Even just saying that short range and long events were observed in the study is probably enough to clarify this.

It has been shown previously that SegA and SegB are involved in chromosome partition, but in addition to advancing understanding of these processes the study now also reveals hints of mechanisms that link chromosome segregation with

chromosome organisation via this system, and more emphasis could perhaps be placed here in the abstract and throughout, as reflected in the title.

In Figure 2A, the graphs look a little more basic and it is unclear if there are any repeats and standard deviations/ statistical analyses for these growth curves?

Why was it not possible to generate SegA KOs but SegA/SegB double KO were generated?

More emphasis could be placed throughout of how cohesin compacts the transcriptionally repressed B-compartment, but here segA-B plays a novel role in compaction in the A -compartment.

Lines 60-62 – perhaps rephrase to something like ‘another key chromosome segregation protein, a eukaryotic condensin orthologue, known as the structural Maintenance of Chromosomes (SMC) protein’ – and SMC superfamily probably needs some introduction

line 164-166 – 2 or 3 instances of ‘structure’ should be corrected to predicted structure, even if the AF model confidence is high!

The AF2 data should probably be repeated using the more recent AF3 release.

Line 169 – it might also be interesting to compare the Dali output with Foldseek analyses?

222 – 15 high and 24 low-enrichment peaks – how were these defined? – perhaps a little more detail here?

342/3 - 49 total sites were identified...via combined in silico and in vitro (Supp Table 1) – can a little more detail be added in the main text?

The AFM results of both local compaction and longer-range DNA-looping are novel and exciting. I do not think further experimental work is necessarily needed, but if the authors were able to perform any further biophysical work on these observations within the review window it might make an even more compelling case for these intriguing new results! Was SegA also investigated in combination with SegB in these AFM studies?

The Model is discussed in final figure legend, but without any expansion in the discussion itself – perhaps the discussion can finish with the putative model.

It is interesting that despite the clear divergence/difference of ParB and SegB in bacteria and archaea, respectively, it seems to be the case that there are perhaps similar modes of compaction and segregation in both systems. This should probably be discussed further and the following paper (and/or others) cited:

Phase-separated ParB enforces diverse DNA compaction modes and stabilizes the parS-centered partition complex
<https://academic.oup.com/nar/article/52/14/8385/7697533>

Overall this is a nice study, which I enjoyed reviewing, with well-executed and broad experimental approaches that advance our understanding of the SegA-SegB system, and in my view the paper should be of sufficient interest to the general readership of Nature Communications.

Reviewer #2

(Remarks to the Author)

In the current research, Kabli et al., investigated the functions of two proteins SegA and SegB in two Sulfolobaceae organisms, *S. acidocaldarius* and *Sa. solfataricus*. Gene deletion strains were constructed for segB and/or segA/B, and ChIP-seq was conducted to identify binding sites of SegB across the genomes of these archaea. In addition, immunofluorescence microscopy was used to identify the localization of SegA and SegB in the cells and AFM was employed to reveal a potential role of SegB in chromosome organization. They found that SegB mainly bound the region surrounding the segAB locus across the chromosome and SegB sites are enriched in the A compartment fraction of the genome in these organisms. Growth and microscopy experiments showed that the segB and/or segAB deletion strains grow more slowly and produce anucleate cells, and AFM suggested a DNA compaction and looping capability for SegB. Together, these results indicate that SegB is involved in chromosome segregation and maybe also in chromosome organization, and thereby establishing an interesting link between chromosome organization and genome segregation. However, since it is not known what roles SegAB could have played even in the archaeal chromosome segregation, and their possible function in chromosome organization remains to be investigated, more works are needed in order to yield insights into the possible connection between SegAB and the linkage of chromosome organization and genome segregation in archaea. In addition, some data need to be confirmed and a part of these results are already known or similar in previous studies (such as the interaction between SegA and SegB, DNA binding motif of SegB). Therefore, this manuscript seems to provide a limited new finding for this field.

Specific points:

1. Line 121-122(Fig. 1): There is an additional gene downstream of *segB* in *Sa. islandicus* compared to that in *Sa. solfataricus*. Does this suggest that the *segAB* locus genomic context is not conserved in this genus?
2. Line 157-167(Supplementary Fig. 4): The level of confidence for beta-sheet in the dimer of *Metallosphaera hakonensis* SegB is low and so is the prediction for the C-terminal domain of *M. hakonensis* SegB monomer, and this raises a question about the validity of these structural predictions. Have the authors tried to AF3 to see if better predictions could be obtained?
3. Have the authors attempted to delete both *segA* and *segB* in *Sa. solfataricus*? If deletable, what is its phenotype? If not, what could be reason why these genes are not essential in *S. acidocaldarius* but essential in *Sa. solfataricus*?
4. If *segB* is essential for chromosome segregation, it should be impossible to obtain any deletion mutant; if obtained, there must be additional mechanisms for chromosome aggregation. The results obtained with Δ *segB* that it hardly grows up under normal condition (Fig. 2A, Supplementary Fig. 5A) support the above reasoning. If so, the authors should discuss alternative segregation mechanisms. If not, it could also be the deletion strain is in fact a hybrid, still carrying wt chromosome. Please experimentally check the genotype of Δ *segB*.
5. Also, it is very strange that both wild type strains of *Sa. solfataricus* and *S. acidocaldarius* grew much worse than previous studies (less than OD~0.5 at 120 h vs OD>1 at 40-60 h) (Baes et al., Extremophile, 2020; Kolk et al., FM, 2020). Is there any problem with the cultivation condition the authors used here? The growth status can radically change the physiology of an organism, including the fractionation of 3-D genome. As reported for *S. acidocaldarius*, its transcriptome profile was significantly remodeled and the extent of chromosome compartmentalization was much weakened in stationary phase compared to those in exponential phase. (Takemata, et al., Cell, 2019.)
6. The microscopy pictures in Fig. 2C seem that the percentage of anucleate cells of wild type *Sa. solfataricus* is higher than that for Δ *segB*, which is quite different from the data in Fig. 2B. In addition, why did *S. acidocaldarius* Δ *segAB* grow much better than Δ *segB*, but had a bit higher percentage of anucleate cells than that for Δ *segB*? Since Δ *segAB* and Δ *segB* contain anucleate cells which indicates a failure in chromosome segregation, did the authors observed the cells with more than 2 copies of chromosomes? Further analysis like flow cytometry would provide more information about the function of SegA/B.
7. Line 261-264: It is not accurate to say that SegB may play a role in chromosome organization just because of several low enrichment of SegB binding sites within the genes of *rpoB2*, *topR2*, and reverse gyrase. Main enriched peaks of SegB were found associated with genes of hypothetical, Vitamin B biosynthesis and transcription regulator. To reveal whether SegB would affect their transcription, ChIP-qPCR and RT-qPCR are suggested to analyze the genes involved in chromosome organization. In addition, it seems very interesting that SegB also located near *oriCs*. Does SegB affect DNA replication?
8. Line 272-273/289-291: The data suggest that the SegB binding motif appears could be only conserved across different genera within Sulfolobaceae or Sulfolobales (not most of archaeal genera mentioned in Abstract) since the authors did not provide data from more species. And the location of the motif in the *Thermococcus sibiricus* shows that it seems randomly distributed at several sites of the genome and not mainly located near SegAB, suggesting that this is not true for Euryarchaea. On the other hand, it may be more reasonable to investigate the matches between SegBs and binding motifs from various species to reveal its conservation and evolution instead of by searching a motif across various genomes.
9. The authors found that SegB sites are enriched in the A compartment fraction of the 3 D-genome in these archaea. Have they performed a statistics analysis on the ratio of SegB binding sites revealed by ChIP-seq to those searched by bioinformatic analysis? This may provide insights into a mechanism involved: the observation that SegB sites are enriched in the A compartment is due to the enrichment of SegB binding sites in the A compartment or preferred binding by SegB on the motifs located in the A compartment.
10. Line 366-367: Since several thermostable fluorescent proteins have been developed, the authors may consider to use one of these tools in the current work. (Recalde et al., FM, 2024)
11. Line 389-392: It is interesting to note that the distribution of SegB foci in the cell may have relationship with its foci numbers. However, the number of cells with one single focus analyzed here is only 19, which is too few to be confirmed (Supplementary Fig.10). If it is the case, immunofluorescence microscopy on the cells after synchronization should be important. This will detect the SegB foci number correlate with their location in the cell during chromosome segregation and cell division, which would provide insight into the potential role of SegB in chromosome organization.
12. *S. acidocaldarius* PCNA also has similar cellular localization (Gristwood et al., NAR, 2012). Is it possible to perform colocalization of PCNA and SegB to detect any linkage of DNA replication and chromosome segregation in Sulfolobales? Or this is just a common feature for the proteins involved in DNA metabolism processes?
13. Line 396: This part is not so informative since the interaction between SegA and SegB had been reported previously (Kalliomaa-Sanford et al., PNAS, 2012; Yen et al., NAR, 2021). The dynamic interaction data in the presence of SegA, SegB and DNA would be more helpful for the manuscript.
14. AFM analysis on several chromatin proteins in Sulfolobales has presented a similar binding profiles on DNA as that in this study (Cajili et al., Biomolecules, 2022; Lemmens et al., Biomolecules, 2022). A control containing a DNA fragment without SegB binding site is need for the AFM experiments to exclude randomly binding by SegB.
15. The foci and AFM experiments of SegB just provide some implications. However, it can not exclude unspecific binding or other functions instead of chromosome organization since investigations on other proteins gave similar results as mentioned above. The role of SegB in chromosome organization should be further investigated by other techniques such as Hi-C analysis on Δ *segB* and/or Δ *segAB*. This would also confirm the function of other SegB binding sites on the chromosome.

Reviewer #3

(Remarks to the Author)

I co-reviewed this manuscript with one of the reviewers who provided the listed reports.

Version 1:

Reviewer comments:

Reviewer #1

(Remarks to the Author)

The authors have addressed my comments satisfactorily and the revised manuscript should now be clearer to readers without background knowledge of the field. The study represents a large, thorough and well-executed body of work that advances our understanding of the SegA/SegB system in archaea and reveals novel insights into the potential roles of these proteins in DNA compaction and chromosome organisation. The single-molecule approaches suggested by the authors in response to my comment relating to further insight into mechanism will no doubt lead to elucidation of the mechanisms involved, and I look forward to seeing this work in a future study!

One minor point:

In response to my initial point "In Figure 2A, the graphs look a little more basic and it is unclear if there are any repeats and standard deviations/ statistical analyses for these growth curves?"

The authors have replied - The growth curves shown in Fig. 2A are representative curves. Three full independent biological experiments were conducted and the repeats are reported in Supplementary Fig. 6.

I would suggest that these replicates should be averaged in the graph in the main figure with the inclusion of error bars, with the three replicates shown in the Supplementary Materials.

In summary, I believe that this modified manuscript is now suitable for publication in Nature Communications.

Reviewer #2

(Remarks to the Author)

The authors have successfully addressed our concerns including structure prediction, strain verification and phenotypic analysis by flow cytometry and included more data in the revised manuscript. In particular, we are glad to see that the flow cytometry data clearly show that all the deletion strains contain polyploid cells, consistent with the role of SegA/B in chromosome segregation. Herein we would like to point out the inconsistency of the locations of the highest peaks in different samples of *Sa. solfataricus*: while WT and Δ segB(psegB) strains show a position of the highest peaks (possibly 2 copies of chromosomes peak) close to 0.1M DNA fluorescence intensity, that of the Δ segB strain is about 0.5M DNA fluorescence intensity. This raises a question whether these highest peaks could all represent cells of 2 chromosomes. If not, the percentage of polyploidy in the segB deletion strain could be much higher than 47.7%. Thus, we recommend accept for this manuscript upon clarifying the X-scale of the *Sa. solfataricus* flow cytometry data.

Reviewer #3

(Remarks to the Author)

[No comments for authors]

Point-by-point response to Reviewers

Manuscript NCOMMS-25-09621

Coupling chromosome organization to genome segregation in Archaea

Kabli *et al.*

We would like to thank the Reviewers and the Editor for reading our manuscript and providing useful comments on different aspects of the work. We have included new data (flow cytometry) and modified some parts of the manuscript to address the queries raised by the Reviewers. Please find a point-by-point response below.

Reviewer 1

This is an interesting study and presents an impressive body of work that significantly advances our understanding of the chromosome segregation processes in Thermoproteota archaea, specifically the SegAB partition system. Crucially, the work provides insights into how these proteins organise and compact chromosomal DNA, seemingly in one of the two major spatial compartments within cells of the Sulfolobales and also highlights interesting putative associations of the compaction/segregation processes with cell division.

The study deploys a broad range of experimental approaches starting with bioinformatics and structural modelling to explore the distribution and likely structures of the SegAB division system. A C-terminal RHH domain in SegB is highlighted and the presence of this domain in other archaeal cell cycle regulator proteins is discussed.

In vivo deletion *segB* and SegAB is shown to cause chromosome segregation and cell division defects, with anucleate and cells displaying aberrant nucleoid morphology, consistent with chromosome segregation defects.

ChIP-seq approaches are then used in several Thermoproteota species to examine the binding localisation of these proteins across these archaeal genomes, and these studies reveal that SegB binds preferentially in the 'A' compartment of the cell that hold the replication origins prior to cell division, and typically the transcriptionally active components. Binding sites are then confirmed in vitro by band-shift and footprinting techniques that reveal lower and high affinity binding sites. The subcellular localisation of the SegA and SegB proteins are also demonstrated by immunofluorescence microscopy, while SegA to SegB binding is confirmed in vitro by microscale thermophoresis MST.

Exciting novel findings were also made towards the end of the study with the in vitro observation of localised DNA compaction following binding at the three Seg B recognition sites at the *segAB* by AFM. These data also provided further evidence of binding to higher and lower affinity sites. Finally, strikingly clear loop formation is demonstrated in vitro by AFM on longer DNA substrates by bridging long-distance sites, suggesting at least two novel DNA compaction modes via SegB in the Sulfolobales.

The article is for the large part well written, but the clarity could perhaps be slightly improved in places for a more generalist audience. For example, the A and B chromosome compartments probably require slightly more introduction. Also, while the abstract is clear enough in general it

might be an idea to detail some of the experimental approaches that lead to the reported discoveries, and importantly there is reference to 'the' Short-range DNA compaction and long term looping – but there is no prior introduction to these events! Even just saying that short range and long events were observed in the study is probably enough to clarify this.

We have worked on improving clarity to make the manuscript more accessible to a generalist audience.

1. We have introduced the A and B chromosome compartments in the Abstract, so that the readers can appreciate our results from the first page of the manuscript. Moreover, more details about the compartments have been added in the Introduction on page 4, where we have also emphasized that while CIsN (coalescin) is responsible for organizing the B compartment, *'to date it remains elusive whether another protein binds and actively establishes the structure of the A compartment.'*
2. We do appreciate that the introduction of the short-range compaction and long-range looping by SegB was slightly sudden, as we were trying to keep the Abstract within the journal's suggested word limit. We have modified the Abstract and introduced those results with the sentence *'Atomic force microscopy experiments provided mechanistic insights into how SegB binds DNA and uncovered short-range DNA compaction and long-range looping of distant sites by SegB'.*

As suggested by the Reviewer, other experimental approaches were also mentioned in the abstract to improve clarity.

It has been shown previously that SegA and SegB are involved in chromosome partition, but in addition to advancing understanding of these processes the study now also reveals hints of mechanisms that link chromosome segregation with chromosome organisation via this system, and more emphasis could perhaps be placed here in the abstract and throughout, as reflected in the title.

We thank the Reviewer for this suggestion. More emphasis on the novel mechanistic aspects of the SegAB system has been placed in the Abstract, Introduction and Discussion.

In Figure 2A, the graphs look a little more basic and it is unclear if there are any repeats and standard deviations/ statistical analyses for these growth curves?

The growth curves shown in Fig. 2A are representative curves. Three full independent biological experiments were conducted and the repeats are reported in Supplementary Fig. 6.

Why was it not possible to generate SegA KOs but SegA/SegB double KO were generated?

We made several attempts to construct the $\Delta segA$ strain, but without success, and at a certain point we decided to move forward the project with the strains that were constructed. Why the construction of $\Delta segAB$ was possible, but $\Delta segA$ was not is an interesting question. The growth curves clearly indicate that the deletion of both genes is better tolerated compared to deletion of a single gene. The co-expression of the two genes from the same promoter indicates that stoichiometric quantities of the proteins are important for their function in the cell. When an imbalance of the two proteins occurs, then problems arise. In bacterial ParAB systems a perfect balance of the two proteins has been reported to be key for the function of the DNA segregation

systems (for example, Mohl and Gober, 1997, *Cell* 5, 675-684; Bartosik *et al*, 2009, *Microbiology* 155, 1080-1092).

This potential explanation has been included in the Discussion on page 21.

More emphasis could be placed throughout of how coalescin compacts the transcriptionally repressed B-compartment, but here segA-B plays a novel role in compaction in the A-compartment.

Thanks for the suggestion. We have been very conscious of the considerable length of the manuscript and have tried to be rather concise in places. As mentioned above, more emphasis on the novel role of SegAB in organizing the A compartment has been placed in the Abstract and Introduction, pointing out that whilst coalescin structures the B compartment, the factor/s responsible for the folding of the A compartment had remained elusive to date.

Lines 60-62 – perhaps rephrase to something like ‘another key chromosome segregation protein, a eukaryotic condensin orthologue, known as the structural Maintenance of Chromosomes (SMC) protein’ – and SMC superfamily probably needs some introduction

The sentence has been rephrased as suggested by the Reviewer and more detail on SMCs has been included (page 3, second paragraph).

line 164-166 – 2 or 3 instances of ‘structure’ should be corrected to predicted structure, even if the AF model confidence is high!

We have modified as suggested (page 6 and 7).

The AF2 data should probably be repeated using the more recent AF3 release.

Thanks for the suggestion. We have repeated the structure predictions with AF2 multimer and carried out predictions using AF3. These recent searches have provided some interesting findings that have been described on page 7 and included in Supplementary Fig. 4.

In summary, one of the models predicted by AF2, which was not selected by the program as the best model, displayed the repeat region of each monomer *M. hakonensis* SegB folded into a β -helix, as observed in the monomer prediction. The two β -helices, one from each monomer, sit next to one another in this predicted dimer, merging into a longer solenoid (Supplementary Fig. 4C, middle). Interestingly, AF3 predicted a very similar structure in which the N-termini of the monomers are folded into adjacent β -helices positioned next to one another, as observed in the AF2-predicted dimer (Supplementary Fig. 4D), although with a slightly higher confidence (predicted Template Modeling, pTM = 0.4; interface predicted Template Modeling, ipTM = 0.35). The predicted C-terminus is folded again with high confidence.

Line 169 – it might also be interesting to compare the Dali output with Foldseek analyses?

Thanks for the suggestion, we have performed Foldseek searches and compared the results with those obtained from the Dali server. The hits are all very similar and the most recurrent ones were the FeS cluster proteins SufB and SufD from different bacteria and archaea (Z-score 18.5 for *E. coli* SufB). The structure of these proteins has been solved and harbours a β -helix

fold (Hirabayashi *et al.* 2015, *J. Biol. Chem.* 290, 29717-29731). Interestingly, *E. coli* SufBD form a heterodimer in which the β -helix of each protein sits next to that of the other protein, resulting into an elongated solenoid structure that closely resembles that of the predicted *M. hakonsensis* SegB dimer (Supplementary Fig. 4E and 4F).

Other hits were cell surface- and membrane-associated proteins.

222 – 15 high and 24 low-enrichment peaks – how were these defined? – perhaps a little more detail here?

We appreciate the point raised by the Reviewer. We had added the detail here originally, but then moved this information to the Materials & Methods. Low-enrichment peaks are defined as peaks with a RPBPM value inferior to 5. We have added this detail in the main text.

342/3 - 49 total sites were identified...via combined in silico and in vitro (Supp Table 1) – can a little more detail be added in the main text?

We have added more detail on page 14. The sentence has been modified to '*Forty-nine total sites were identified on the S. solfataricus chromosome: the bioinformatic analysis identified forty-five sites. Moreover, DNase I footprints detected four additional sites, bringing the total number of identified sites to forty-nine (Supplementary Table S1)*'.

The AFM results of both local compaction and longer-range DNA-looping are novel and exciting. I do not think further experimental work is necessarily needed, but if the authors were able to perform any further biophysical work on these observations within the review window it might make an even more compelling case for these intriguing new results! Was SegA also investigated in combination with SegB in these AFM studies?

Thanks for the kind words of appreciation. We plan to perform further biophysical experiments to investigate DNA condensation and looping by SegB, for example by using optical tweezers with immobilised DNA to observe DNA shortening. We have to establish appropriate collaborations and identify core facilities where we can carry out the experiments. Thus, this will be a future objective.

We have not performed AFM experiments with both proteins as yet.

The Model is discussed in final figure legend, but without any expansion in the discussion itself – perhaps the discussion can finish with the putative model.

Thanks for the suggestion, we have now concluded the discussion with a description of the proposed model.

It is interesting that despite the clear divergence/difference of ParB and SegB in bacteria and archaea, respectively, it seems to be the case that there are perhaps similar modes of compaction and segregation in both systems. This should probably be discussed further and the following paper (and/or others) cited:

Phase-separated ParB enforces diverse DNA compaction modes and stabilizes the parS-

centered partition complex

<https://academic.oup.com/nar/article/52/14/8385/7697533>

Thanks for the comments. Indeed, it is interesting that different domains of life deploy evolutionarily unrelated DNA binding proteins to mediate chromosome compaction and segregation possibly using analogous mechanisms and principles. We have included this observation in the Discussion (page 22) and cited the suggested paper together with others.

Overall this is a nice study, which I enjoyed reviewing, with well-executed and broad experimental approaches that advance our understanding of the SegA-SegB system, and in my view the paper should be of sufficient interest to the general readership of Nature Communications.

Thanks for the supportive comments.

Reviewers 2 and 3

In the current research, Kabli et al., investigated the functions of two proteins SegA and SegB in two Sulfolobaceae organisms, *S. acidocaldarius* and *Sa. solfataricus*. Gene deletion strains were constructed for *segB* and/or *segA/B*, and ChIP-seq was conducted to identify binding sites of SegB across the genomes of these archaea. In addition, immunofluorescence microscopy was used to identify the localization of SegA and SegB in the cells and AFM was employed to reveal a potential role of SegB in chromosome organization. They found that SegB mainly bound the region surrounding the *segAB* locus across the chromosome and SegB sites are enriched in the A compartment fraction of the genome in these organisms. Growth and microscopy experiments showed that the *segB* and/or *segAB* deletion strains grow more slowly and produce anucleate cells, and AFM suggested a DNA compaction and looping capability for SegB. Together, these results indicate that SegB is involved in chromosome segregation and maybe also in chromosome organization, and thereby establishing an interesting link between chromosome organization and genome segregation. However, since it is not known what roles SegAB could have played even in the archaeal chromosome segregation, and their possible function in chromosome organization remains to be investigated, more works are needed in order to yield insights into the possible connection between SegAB and the linkage of chromosome organization and genome segregation in archaea. In addition, some data need to be confirmed and a part of these results are already known or similar in previous studies (such as the interaction between SegA and SegB, DNA binding motif of SegB). Therefore, this manuscript seems to provide a limited new finding for this field.

Specific points:

1. Line 121-122(Fig. 1): There is an additional gene downstream of *segB* in *Sa. islandicus* compared to that in *Sa. solfataricus*. Does this suggest that the *segAB* locus genomic context is not conserved in this genus?

The extra gene present downstream of *segB* in the chromosome of *S. islandicus* is a frameshifted pseudogene encoding a transposase. All the other genes upstream and downstream of the *segAB* operon are entirely conserved, indicating that the genomic neighbourhood is highly conserved in the *Saccharolobus* genus.

2. Line 157-167(Supplementary Fig. 4): The level of confidence for beta-sheet in the dimer of *Metallosphaera hakonensis* SegB is low and so is the prediction for the C-terminal domain of *M.*

hakonensis SegB monomer, and this raises a question about the validity of these structural predictions. Have the authors tried to AF3 to see if better predictions could be obtained?

AlphaFold is an exploratory tool to get insights into potential structures. As stated in the manuscript, we agree with the Reviewers that the level of confidence for the β -sheet in the *Metallosphaera hakonensis* SegB dimer is low, but the prediction remains tantalizing and as such deserves further investigation.

Thanks for the AF3 suggestion. We have repeated the structure predictions with AF2 multimer and carried out predictions using AF3. These recent searches have provided some interesting findings that have been described on page 7 and included in Supplementary Fig. 4. The C-terminus of *M. hakonensis* SegB monomer is now predicted with higher confidence (Supplementary Fig. 4B).

In summary, one of the models predicted by AF2 multimer, which was not selected by the program as the best model, displayed the repeat region of each monomer *M. hakonensis* SegB folded into a β -helix as observed in the monomer prediction. The two β -helices, one from each monomer, sit next to one another in this predicted dimer, merging into a longer solenoid (Supplementary Fig. 4C, middle). Interestingly, AF3 predicted a very similar structure in which the N-termini of the monomers are folded into adjacent β -helices positioned next to one another, as observed in the AF2-predicted dimer (Supplementary Fig. 4D), although with a slightly higher confidence (predicted Template Modeling, pTM = 0.4; interface predicted Template Modeling, ipTM = 0.35). The predicted C-terminus is folded again with high confidence.

3. Have the authors attempted to delete both *segA* and *segB* in *Sa. solfataricus*? If deletable, what is its phenotype? If not, what could be reason why these genes are not essential in *S. acidocaldarius* but essential in *Sa. solfataricus*?

Yes, we have tried to construct a *S. solfataricus* double *segAB* deletion, but without success. We made five attempts and then decided to proceed with only $\Delta segB$ to progress with the project.

At this stage I would say that we do not know for sure whether both *segA* and *segB* are essential in *S. solfataricus*. The *segB* gene appears not to be essential, although the cells are

clearly unhealthy and struggle to grow. If *segAB* were proven essential genes in *S. solfataricus*, then the difference between *S. solfataricus* and *S. acidocaldarius* might be explained considering that the two organisms belong to two different genera and have DNA segregation machines that may function slightly differently. For example, upstream of *segAB* in *S. solfataricus* there is a gene that we called *segC*, which encodes for a protein that interacts with the SegAB complex and forms large, textured filaments *in vitro* (Lin *et al*, 2024, *Nucleic Acid Res* 52, 9966-9977). This protein is not present upstream of *segAB* in *S. acidocaldarius*.

Another example highlighting cell cycle differences between the *Sulfolobus* and the *Saccharolobus* genera was recently provided by a cell division study by the Krupovic group. These investigations showed that '*ESCRT-III undergoes a relatively slow degradation in S. islandicus, contrasting the situation in S. acidocaldarius, where rapid degradation of CdvB (ESCRT-III) initiates the membrane constriction by CdvB1 (ESCRT-III-1) and CdvB2 (ESCRT-III-1)*' (Liu *et al*, 2024, *mBio* 16: e00991-24).

More broadly, it is useful to mention here that in bacteria the *parAB* system is essential for viability only in some species, such as *Caulobacter crescentus* (Mohl and Gober, 1997, *Cell* 5, 675-684). However, even when the system is not strictly required for viability, deletion of the genes results in a significant increase in anucleate cells, substantiating the important role played in chromosome segregation (Badrinarayanan *et al*, 2015, *Annu Rev Cell Dev Biol* 31, 171-199).

Some of these comments have been included in the Discussion (page 20), as they might be of interest to a general readership.

4. If *segB* is essential for chromosome segregation, it should be impossible to obtain any deletion mutant; if obtained, there must be additional mechanisms for chromosome aggregation. The results obtained with Δ *segB* that it hardly grows up under normal condition (Fig. 2A, Supplementary Fig. 5A) support the above reasoning. If so, the authors should discuss alternative segregation mechanisms. If not, it could also be the deletion strain is in fact a hybrid, still carrying wt chromosome. Please experimentally check the genotype of Δ *segB*.

We were able to delete *segB* in both *S. solfataricus* and *S. acidocaldarius*, indicating that the gene is not essential for viability. However, the gene encodes a key player for chromosome segregation and the absence results in anucleate cells. Indeed, we agree that there must be a level of redundancy and multiple systems/mechanisms might be responsible for chromosome segregation.

We exclude the possibility that the Δ *segB* deletion strain is a hybrid containing both a wild type chromosome and a Δ *segB* chromosome. We have provided evidence that is shown in Supplementary Fig. 5. Moreover, all the deletion strains had been subjected to whole genome sequencing showing that each contains only the chromosome without the target gene. These results have been reported on page 8 of the Results.

5. Also, it is very strange that both wild type strains of *Sa. solfataricus* and *S. acidocaldarius* grew much worse than previous studies (less than OD~0.5 at 120 h vs OD>1 at 40-60 h) (Baes *et al.*, *Extremophile*, 2020; Kolk *et al.*, *FM*, 2020). Is there any problem with the cultivation condition the authors used here? The growth status can radically change the physiology of an organism, including the fractionation of 3-D genome. As reported for *S. acidocaldarius*, its transcriptome profile was significantly remodeled and the extent of chromosome

compartmentalization was much weakened in stationary phase compared to those in exponential phase. (Takemata, et al., Cell, 2019.)

The works by Baes *et al* and Kolk *et al* utilised *S. acidocaldarius*. The growth rate of cultures will always show some variability in different laboratories and depends on the initial inoculum and growth medium.

As for the *S. solfataricus* culture, the strain used is PBL2025 that is strain 98/2 $\Delta(sso3004-3050)$ (Schelert *et al*, 2004, *J Bacteriol* 186, 427-437). This strain presents a 58 kbp deletion in the chromosome which might affect its growth as compared to wild type strains such as P2. The PBL2025 strain was used to construct the *S. solfataricus* $\Delta segB$ deletion strain as PBL2025 lacks *lacS* and this selectable marker replaced the *segB* gene.

Our laboratory has grown *S. solfataricus* and *S. acidocaldarius* cultures for almost two decades and I believe we are experienced in handling these organisms. We would like to reassure the Reviewers that the cultures used for ChIP-seq were all exponential (as noted in the Methods) and healthy-growing.

6. The microscopy pictures in Fig. 2C seem that the percentage of anucleate cells of wild type *Sa. solfataricus* is higher than that for $\Delta segB$, which is quite different from the data in Fig. 2B.

We appreciate that the wild type image contrast might not be the most optimal. However, almost the totality of the wild type cells shown in Fig. 2C have the nucleoid. We would like to point out that in many of the wild type cells the DAPI-stained nucleoid assumes a crescent shape morphology, appearing as a thin arc squeezed against the cell membrane (see example cell below). This phenomenon has been previously reported (Poplawski & Bernander, 1997, *J Bacteriol* 179, 7625-7630; Takemata *et al*, 2019, *Cell* 179, 165-179). The chromosome appears instead more diffuse in the $\Delta segB$ strain.

Some of these comments have been included in the Results (page 9), as these features might be of interest to a general readership.

In addition, why did *S. acidocaldarius* $\Delta segAB$ grow much better than $\Delta segB$, but had a bit higher percentage of anucleate cells than that for $\Delta segB$?

Indeed, as the growth curves show (Fig. 2A), the *S. acidocaldarius* $\Delta segAB$ strain has a better fitness than the single $\Delta segB$ deletion strain. It appears that the deletion of both genes is less deleterious and better tolerated than the deletion of *segB* alone, despite the marginally higher percentage of anucleate cells. However, the co-expression of the two genes from the same promoter indicates that stoichiometric quantities of the proteins are important for their function in the cell. When an imbalance of the two proteins occurs, then problems arise. In bacterial ParAB systems a perfect balance of the two proteins has been reported to be key for the function of the DNA segregation systems (for example, Mohl and Gober, 1997, *Cell* 5, 675-684;

Bartosik *et al*, 2009, *Microbiology* 155, 1080-1092).

Since Δ segAB and Δ segB contain anucleate cells which indicates a failure in chromosome segregation, did the authors observed the cells with more than 2 copies of chromosomes? Further analysis like flow cytometry would provide more information about the function of SegA/B.

Thanks for the useful suggestion. We have performed flow cytometry experiments and the results have been included in the manuscript (Supplementary Fig. 7 and 8). This approach has shown that all the deletion strains contain polyploid cells. The level of polyploid cells is particularly high in the *S. solfataricus* Δ segB strain (47.7 %) (Supplementary Fig. 7). Interestingly, upon complementation by introduction of *segB* on a plasmid, the polyploidy of the strain returns to wild type level.

Interestingly, the flow cytometry analysis has also revealed that *S. solfataricus* Δ segB displays a more heterogeneous population characterized by a broader range of cell sizes and complexities reflective of polyploidy compared to the wild type strain (Supplementary Fig. 8).

7. Line 261-264: It is not accurate to say that SegB may play a role in chromosome organization just because of several low enrichment of SegB binding sites within the genes of *rpoB2*, *topR2*, and reverse gyrase. Main enriched peaks of SegB were found associated with genes of hypothetical, Vitamin B biosynthesis and transcription regulator. To reveal whether SegB would affect their transcription, ChIP-qPCR and RT-qPCR are suggested to analyze the genes involved in chromosome organization. In addition, it seems very interesting that SegB also located near oriCs. Does SegB affect DNA replication?

The ChIP-seq analysis has revealed a substantial enrichment peak (peak E) (Fig. 3) within the gene encoding for coalescin ClsN that has an established role in chromosome compartmentalization (Takemata *et al*, 2019, *Cell* 179, 165-179).

Moreover, additional low-enrichment peaks were found within the gene SSO_RS01130 (*rpoB2*) and SSO_RS04795 (*topR-2*) that encode the B" subunit of RNA polymerase and the reverse gyrase, respectively, in *S. solfataricus*. The common denominator for these proteins is the involvement in chromosome organization, folding and topology. We feel that this is an exciting connection and agree that it deserves further investigation in the future.

At this stage, we do not know whether SegB has a role in DNA replication. This might be an interesting lead for future investigations, but is beyond the scope of this work.

8. Line 272-273/289-291: The data suggest that the SegB binding motif appears could be only conserved across different genera within Sulfolobaceae or Sulfolobales (not most of archaeal genera mentioned in Abstract) since the authors did not provide data from more species. And the location of the motif in the *Thermococcus sibiricus* shows that it seems randomly distributed at several sites of the genome and not mainly located near SegAB, suggesting that this is not true for Euryarchaea. On the other hand, it may be more reasonable to investigate the matches between SegBs and binding motifs from various species to reveal its conservation and evolution instead of by searching a motif across various genomes.

Conservation of the binding motif in Sulfolobaceae. In the abstract we state that the binding sites are localised close to *segAB* in most archaeal genera. We have changed the sentence to

'ChIP-seq investigations revealed that SegB binds to multiple sites scattered across the chromosome, but mainly localised close to the segAB locus in most of the examined archaeal genera'.

As for *Thermococcus sibiricus*, the bioinformatic identification of fewer SegB binding sites clearly suggests that the motif might be more degenerate in Euryarchaeota and, hence, more challenging to be discovered. We would not say that the sites are *'randomly distributed'*, as the seven sites were identified with a high *P* value (between 6.6 and 8.50 1E-06) and using a *P* threshold value of 1.0E-5. The consensus sequence of the identified sites is very similar to that of *S. solfataricus* (TCT AGA a/t TCT ACA). It is likely that other sites that harbour a sequence quite different from the motif observed in Sulfolobaceae are present in proximity to the *segAB* genes and elsewhere on the chromosome. To test this hypothesis, the *P* value threshold for the search was lowered (from 1.0E-5 to 1.0E-4) and 193 sequences were returned, indicating that SegB sites are more degenerate in *Thermococcus sibiricus* (page 12).

Our prediction is that other sites harbouring a sequence quite different from the motif observed in Sulfolobaceae are present in proximity to the *segAB* genes. This is a general rule for *parAB* systems on bacterial chromosomes.

We are not entirely sure as to what the Reviewers mean with *'to investigate the matches between SegBs and binding motifs from various species to reveal its conservation and evolution'*. The experimental verification of the binding of different SegB orthologues to their cognate DNA motifs is beyond the scope of this work and perhaps a project for the future.

9. The authors found that SegB sites are enriched in the A compartment fraction of the 3 D-genome in these archaea. Have they performed a statistics analysis on the ratio of SegB binding sites revealed by ChIP-seq to those searched by bioinformatic analysis? This may provide insights into a mechanism involved: the observation that SegB sites are enriched in the A compartment is due to the enrichment of SegB binding sites in the A compartment or preferred binding by SegB on the motifs located in the A compartment.

We sincerely appreciate the Reviewers' suggestion that aims to identify the mechanism through which SegB sites are enriched in the A compartment. However, as we state in the manuscript (page 13), the SegB binding motif identified bioinformatically was found in ALL the sequences of the 32 ChIP-seq peaks. The bioinformatic analysis of the chromosome revealed 34 individual sites and the ChIP-seq experiment revealed 32 peaks, each containing one SegB site (Supplementary Table S2). Later we identified 3 additional sites by DNase I footprint in the sequence corresponding to peak 15, bringing the total number of experimentally identified sites to **35** (page 15). The numbers speak clearly in support of the 1st hypothesis outlined above by the Reviewers: the numbers support an enrichment of SegB in the A compartment, because 29 out of 35 binding sites are located in the A compartment and only 6 are in the B compartment, linking SegB function and territory of action to the A compartment of the chromosome.

10. Line 366-367: Since several thermostable fluorescent proteins have been developed, the authors may consider to use one of these tools in the current work. (Recalde et al., FM, 2024)

I appreciate the Reviewers' suggestion. it will be interesting to explore these new tools. However, this is a good project for future investigations.

11. Line 389-392: It is interesting to note that the distribution of SegB foci in the cell may have relationship with its foci numbers. However, the number of cells with one single focus analyzed

here is only 19, which is too few to be confirmed (Supplementary Fig.10). If it is the case, immunofluorescence microscopy on the cells after synchronization should be important. This will detect the SegB foci number correlate with their location in the cell during chromosome segregation and cell division, which would provide insight into the potential role of SegB in chromosome organization.

It is useful to clarify here that we analysed 1026 cells, which we consider a reasonably robust n number for statistics. Out of 1026, 19 cells presented a single SegB focus or patch (Fig. 7E and Supplementary Fig. 10A). This observation reflects the result that SegB binds site-specifically to multiple sites across the chromosome, forming multiple foci in the large majority of cells. The presence of a single large focus or patch is likely to result from the coalescence of smaller SegB foci. The bottom line here is that even if we carry out immunofluorescence microscopy on synchronized cells, the number of cells with a single large focus will still be low, as the most prevalent scenario is ~4-6 foci per cell.

Studying the localisation of SegB in synchronized cell will be an objective for future studies.

12. *S. acidocaldarius* PCNA also has similar cellular localization (Gristwood et al., NAR, 2012). Is it possible to perform colocalization of PCNA and SegB to detect any linkage of DNA replication and chromosome segregation in Sulfolobales? Or this is just a common feature for the proteins involved in DNA metabolism processes?

The analogy suggested by the Reviewers between PCNA and SegB localization in the cell is indeed interesting, with particular reference to the presence of multiple foci and the biased localization at the cell periphery. Potential interactions of protein machineries with the cell membrane are extremely interesting. A potential connection between chromosome replication and segregation is super interesting, but beyond the scope of this work.

13. Line 396: This part is not so informative since the interaction between SegA and SegB had been reported previously (Kallioma-Sanford et al., PNAS, 2012; Yen et al., NAR, 2021). The dynamic interaction data in the presence of SegA, SegB and DNA would be more helpful for the manuscript.

We respectfully disagree with the Reviewers on this point. Previous published work from our lab has presented qualitative results from pull-down and cross-linking experiments showing the interaction of SegA with SegB. However, in this manuscript we have quantified the strength of the interaction and determined the K_D . This value reveals that the affinity between the two proteins is high, which is a critical piece of information in formulating a model of how the SegAB system works.

14. AFM analysis on several chromatin proteins in Sulfolobales has presented a similar binding profiles on DNA as that in this study (Cajili et al., Biomolecules, 2022; Lemmens et al., Biomolecules, 2022). A control containing a DNA fragment without SegB binding site is need for the AFM experiments to exclude randomly binding by SegB.

The observation raised by the Reviewers that *Sulfolobus* proteins involved in chromosome compaction such as Alba and Cren7 form DNA structures *in vitro* similar to those observed for SegB provides further support to the role of SegB in DNA compaction.

However, SegB is a sequence-specific DNA binding protein, whereas Alba and Cren7 bind DNA without sequence specificity. At the concentrations used in the AFM experiments (700 nM or lower), SegB does not exhibit any non-specific DNA binding. We tested this aspect in the past using an unrelated DNA fragment (Kalliomaa-Sanford *et al*, 2012, *Proc Natl Acad Sci USA* 109, 3754-3759, see Fig. 2B). Please also see the answer to the next point.

15. The foci and AFM experiments of SegB just provide some implications. However, it can not exclude unspecific binding or other functions instead of chromosome organization since investigations on other proteins gave similar results as mentioned above. The role of SegB in chromosome organization should be further investigated by other techniques such as Hi-C analysis on Δ segB and/or Δ segAB. This would also confirm the function of other SegB binding sites on the chromosome.

Point on non-specific DNA binding

We can exclude that the foci are due to non-site-specific DNA binding by SegB. For this experiment we have used a 289 bp DNA region that we had previously extensively investigated by deploying multiple DNA-protein interaction techniques. Specifically, DNase I footprinting had compellingly shown site-specific binding of the cognate three sites by SegB (Kalliomaa-Sanford *et al*, 2012, *Proc Natl Acad Sci USA* 109, 3754-3759). The DNA binding pattern observed with AFM in Fig. 8B mirrors exactly what was observed with DNase I footprint, i. e. progressive binding of the three sites by SegB. Furthermore, the foci are formed exactly where the SegB cognate sites are located.

Point on function

The AFM experiments (Fig. 8) show not only that SegB binds DNA site-specifically, but the statistical analysis of the complexes' contour length shows also that SegB binding causes DNA compaction, as the DNA contour length becomes progressively shorter with increasing concentrations of the protein. This result provides evidence that SegB has a role in DNA condensation. Based on these findings, what we state in the manuscript is '*Given the number of SegB sites scattered across the chromosome, the cumulative effect of this short-range activity may play an important role in chromosome condensation in all archaeal genera that deploy SegB orthologues.*'

Furthermore, previous data from our and collaborators' group showed that SegB binding mediates packaging of DNA into a chromatin-like fiber (Yen *et al*, 2021, *Nucleic Acid Res* 49, 13150-13164).

The most obvious alternative is that SegB may be a global transcriptional regulator. However, the presence of the sites almost exclusively in intragenic regions does not lend support to this alternative hypothesis.

We agree with the Reviewers in that Hi-C would provide further evidence in support of the role of SegB in chromosome organization. We plan to perform this analysis in the future, as it is not a trivial method to be set up from scratch.

Point-by-point response to Reviewers

Manuscript NCOMMS-25-09621A

Coupling chromosome organization to genome segregation in Archaea

Kabli *et al.*

We thank the reviewers for making the time to assess the revised version of our manuscript and for their comments.

Reviewer #1

The authors have addressed my comments satisfactorily and the revised manuscript should now be clearer to readers without background knowledge of the field. The study represents a large, thorough and well-executed body of work that advances our understanding of the SegA/SegB system in archaea and reveals novel insights into the potential roles of these proteins in DNA compaction and chromosome organisation. The single-molecule approaches suggested by the authors in response to my comment relating to further insight into mechanism will no doubt lead to elucidation of the mechanisms involved, and I look forward to seeing this work in a future study!

One minor point:

In response to my initial point "In Figure 2A, the graphs look a little more basic and it is unclear if there are any repeats and standard deviations/ statistical analyses for these growth curves?"

The authors have replied - The growth curves shown in Fig. 2A are representative curves. Three full independent biological experiments were conducted and the repeats are reported in Supplementary Fig. 6.

I would suggest that these replicates should be averaged in the graph in the main figure with the inclusion of error bars, with the three replicates shown in the Supplementary Materials.

In summary, I believe that this modified manuscript is now suitable for publication in Nature Communications.

We thank the Reviewer for the kind words. All the replicates of the growth curve experiments have now been plotted together and presented in Fig. 2a with error bars.

Reviewer #2

The authors have successfully addressed our concerns including structure prediction, strain verification and phenotypic analysis by flow cytometry and included more data in the revised manuscript. In particular, we are glad to see that the flow cytometry data clearly show that all the deletion strains contain polyploid cells, consistent with the role of SegA/B in chromosome segregation. Herein we would like to point out the inconsistency of the locations of the highest peaks in different samples of *Sa. solfataricus*: while WT and Δ segB(psegB) strains show a position of the highest peaks (possibly 2 copies of chromosomes peak) close to 0.1M DNA fluorescence intensity, that of the Δ segB strain is about 0.5M DNA fluorescence intensity. This raises a question whether these highest peaks could all represent cells of 2 chromosomes. If not, the percentage of polyploidy in the segB deletion strain could be much higher than 47.7%. Thus, we recommend accept for this manuscript upon clarifying the X-scale of the *Sa. solfataricus* flow cytometry data.

We thank Reviewer 2 & 3 for identifying the inconsistency in *S. solfataricus* flow cytometry data. To clarify this issue, we re-checked the data, metadata and settings for the experiments and realized that the data selected for WT and $\Delta segB(psegB)$ had been acquired at a lower gain compared to that used for the $\Delta segB$ cells.

To address the issue, we have selected comparable flow cytometry experiments for WT and $\Delta segB(psegB)$ cells that were acquired with the same gain as $\Delta segB$ and re-analysed all the data (WT, $\Delta segB$ and $\Delta segB(psegB)$) applying exactly the same gating. The results have been re-plotted in Supplementary Fig. 6 and 7. The histograms and dot plots now show that the prominent 2N peak is positioned at ~ 0.5 M on the fluorescence X axis for all the strains. In the revised experiments, the WT strain shows on average 26% of polyploid cells against 45% of the deletion strain. The expression of *segB* from the plasmid in the $\Delta segB(psegB)$ rescues the defect, reducing the polyploidy to a level even lower than that exhibited by the WT strain. The main manuscript text has been modified accordingly.